# Physiological and motion signatures in static and time-varying functional connectivity and their subject identifiability

**Alba Xifra-Porxas[1][†]\*, Michalis Kassinopoulos[1][†], Georgios D Mitsis[2]\***

[1]Graduate Program in Biological and Biomedical Engineering, McGill University, Montréal, Canada; [2]Bioengineering Department, McGill University, Montréal, Canada

**Abstract** Human brain connectivity yields significant potential as a noninvasive biomarker. Several studies have used fMRI-based connectivity fingerprinting to characterize individual patterns of brain activity. However, it is not clear whether these patterns mainly reflect neural activity or the effect of physiological and motion processes. To answer this question, we capitalize on a large data sample from the Human Connectome Project and rigorously investigate the contribution of the aforementioned processes on functional connectivity (FC) and time-varying FC, as well as their contribution to subject identifiability. We find that head motion, as well as heart rate and breathing fluctuations, induce artifactual connectivity within distinct resting-state networks and that they correlate with recurrent patterns in time-varying FC. Even though the spatiotemporal signatures of these processes yield above-chance levels in subject identifiability, removing their effects at the preprocessing stage improves identifiability, suggesting a neural component underpinning the inter-individual differences in connectivity.

**\*For correspondence:**
alba.xifraporxas@mail.mcgill.ca
(AX-P);
georgios.mitsis@mcgill.ca (GDM)

[†]These authors contributed equally to this work

**Competing interests:** The authors declare that no competing interests exist.

## Introduction

Functional magnetic resonance imaging (fMRI) is based on the blood-oxygen-level-dependent (BOLD) contrast mechanism (*Ogawa et al., 1990*), and is widely viewed as the gold standard for studying brain function because of its high spatial resolution and non-invasive nature. The BOLD signal exhibits low frequency (~0.01–0.15 Hz) fluctuations that are synchronized across different regions of the brain, a phenomenon known as functional connectivity (FC). FC has been observed even in the absence of any explicit stimulus or task, giving rise to the so-called resting-state networks (RSNs) (*Biswal et al., 1995*; *Fox and Raichle, 2007*; *Smith et al., 2009*). Initially, FC was viewed as a stationary phenomenon (static FC) and was commonly measured as the correlation between brain regions over an entire scan. However, several researchers challenged this assumption (*Chang and Glover, 2010*; *Sakoğlu et al., 2010*), and recent studies have been focusing on FC dynamics, quantified over shorter time scales than the scan duration (time-varying FC) (*Hutchison et al., 2013*; *Lurie et al., 2020*).

Although the neurophysiological basis of resting-state FC measured with fMRI is not yet fully understood, many studies have provided evidence to support its neuronal origin. For instance, in animal models, a strong association between spontaneous BOLD fluctuations and neural activity, in particular band-limited local field potentials and firing rates, has been reported (*Logothetis et al., 2001*; *Schölvinck et al., 2010*; *Shmuel and Leopold, 2008*; *Thompson et al., 2013b*). Furthermore, a recent study suggested a close correspondence between windowed FC calculated from simultaneously recorded hemodynamic signals and calcium transients (*Matsui et al., 2019*). In human

studies, direct measurements of macroscale neural activity have revealed a spatial correlation structure similar to that of spontaneous BOLD fluctuations (*Brookes et al., 2011*; *Hacker et al., 2017*; *He et al., 2008*; *Hipp et al., 2012*; *Kucyi et al., 2018*), even during transient (50–200 ms) events (*Baker et al., 2014a*; *Hunyadi et al., 2019*; *Vidaurre et al., 2018*). Therefore, it is widely assumed that resting-state FC measured using BOLD fMRI reflects spontaneous co-fluctuations of the underlying neuronal networks.

However, BOLD signals rely on changes in local cerebral blood flow (CBF) to infer underlying changes in neuronal activity, and according to a recent study, at least 50% of the spontaneous hemodynamic signal is unrelated to ongoing neural activity (*Winder et al., 2017*). For instance, systemic physiological functions can induce variations in global and local CBF, which in turn result in BOLD signal fluctuations. In particular, low frequency variations in breathing activity (*Birn et al., 2008b*; *Birn et al., 2006*; *Power et al., 2017*), arterial blood pressure (*Whittaker et al., 2019*), arterial $CO_2$ concentration (*Prokopiou et al., 2019*; *Wise et al., 2004*), and heart rate (*Chang et al., 2009*; *Shmueli et al., 2007*) are known to account for a considerable fraction of variance of the BOLD signal, presumably through changes in CBF. In addition, the BOLD signal intensity is distorted by high-frequency physiological fluctuations, such as cardiovascular pulsation and breathing, through displacement of the brain tissues and perturbations of the $B_0$ magnetic field (*Dagli et al., 1999*; *Glover et al., 2000*). Further, head motion is well-known to have a substantial impact on fMRI through partial volume, magnetic inhomogeneity and spin-history effects (*Friston et al., 1996*; *Power et al., 2012*). These non-neuronal factors may introduce common variance components in signals recorded from different brain regions and subsequently induce spurious correlations between these areas (*Chen et al., 2020*). Therefore, to account for motion-related and physiological confounds, nuisance regressors are typically obtained using model-based and data-driven techniques, and regressed out from the fMRI data before further analysis (*Caballero-Gaudes and Reynolds, 2017*).

Static and time-varying resting-state FC have shown promise for providing concise descriptions of how the brain changes across the lifespan (*Battaglia et al., 2020*; *Chan et al., 2014*; *Ferreira et al., 2016*; *Geerligs et al., 2015*; *Sala-Llonch et al., 2015*; *Xia et al., 2019*), and to assay neural differences that are associated with disease (*Baker et al., 2014b*; *Chen et al., 2017*; *Damaraju et al., 2014*; *Demirtaş et al., 2016*; *Drysdale et al., 2017*; *Du et al., 2016*; *Gratton et al., 2019*; *Hahamy et al., 2015*; *Mash et al., 2019*; *Morgan et al., 2017*; *Xia et al., 2018*). However, recent studies assessing the performance of a large range of preprocessing strategies found that there is always a trade-off between adequately removing confounds from fMRI data and preserving the signal of interest (*Ciric et al., 2017*; *Kassinopoulos and Mitsis, 2021a*; *Parkes et al., 2018*). Importantly, these studies found that widely used techniques for the preprocessing of fMRI data may not efficiently remove physiological and motion artifacts. The latter raises a concern, as it is still not clear how nuisance fluctuations may impact the outcome of FC studies.

Several studies have examined whether physiological fluctuations across the brain could give rise to structured spatial patterns that resemble common RSNs, based on the evidence that vascular responses following systemic changes are spatially heterogenous (*Chang et al., 2009*; *Pinto et al., 2017*), or account for the observed time-varying interactions between RSNs. For instance, *Bright and Murphy, 2015* applied independent component analysis (ICA) to the fraction of the fMRI data explained by nuisance regressors related to head motion and physiological variability and revealed a characteristic network structure similar to previously reported RSNs. Similarly, *Tong et al., 2013*; *Tong and Frederick, 2014* found significant contributions of systemic fluctuations on ICA time courses related to the visual, sensorimotor and auditory networks. Recently, *Chen et al., 2020* generated BOLD data containing only slow respiratory-related dynamics and showed that respiratory variation can give rise to apparent neurally-related connectivity patterns. Further, recent investigations have shown that physiological confounds can modulate time-varying FC measures (*Chang et al., 2013*; *Nalci et al., 2019*; *Nikolaou et al., 2016*). These results suggest that changes in brain physiology, breathing patterns, heart rhythms and head motion across sessions, within-subject or across populations, may introduce artifactual inter-individual and group-related differences in FC independent of any underlying differences in neural activity. For instance, cardiac autonomic dysregulation has been associated to a variety of psychiatric disorders (*Alvares et al., 2016*; *Benjamin et al., 2020*), which could in principle lead to group differences in connectivity patterns between patients and controls if the effects of heart rate are not accounted

for. Therefore, the disentanglement of the neural and physiological correlates of resting-state FC is crucial for maximizing its clinical impact.

While the previous findings provide evidence for the dual-nature of RSNs in both static and time-varying scenarios, only specific physiological processes and/or particular brain networks were evaluated in each of the aforementioned studies. A more holistic assessment of the impact of these non-neural processes on FC measures is needed to better understand whether and how systemic fluctuations, as well as head or breathing motion affect inter-individual and group differences. Importantly, the wide range of possible preprocessing strategies needs to be reassessed taking into consideration the effects of several non-neural processes on FC measures rather than accounting only for the effects of a specific process (e.g. head motion).

The varying efficiency of different preprocessing pipelines with respect to removing the effect of physiological fluctuations and motion has also implications for studies investigating properties of FC at the individual level. Recent studies have shown that connectivity profiles vary substantially between individuals, acting as an identifying fingerprint (*Finn et al., 2015*; *Miranda-Dominguez et al., 2014*) that is stable over long periods of time (*Horien et al., 2019*). However, the high subject discriminability of connectivity profiles may arise partly as a result of physiological processes (*Batchvarov et al., 2002*; *Golestani et al., 2015*; *Malik et al., 2008*; *Pinna et al., 2007*; *Pitzalis et al., 1996*; *Power et al., 2020*; *Reland et al., 2005*) and head motion (*Van Dijk et al., 2012*; *Zeng et al., 2014*) being highly subject-specific. Evidence supporting this hypothesis comes from studies showing that the mean of intraclass correlation values of functional connections, associated to test-retest reliability across sessions, is reduced when a relatively aggressive pipeline is used (*Birn et al., 2014*; *Kassinopoulos and Mitsis, 2021a*; *Parkes et al., 2018*), suggesting that artifacts exhibit high subject specificity. However, the relation between subject discrimination and inter-individual differences in physiological processes and head motion has yet to be addressed.

In the present work, we capitalize on 3T resting-state fMRI data from the Human Connectome Project (HCP) to uncover the whole-brain connectome profiles of systemic low-frequency oscillations (SLFOs) associated with heart rate and breathing patterns, cardiac pulsatility, breathing motion and head motion on estimates of static and time-varying FC. To quantify the contributions of physiological processes and head motion on FC, we employ model-based techniques with externally recorded physiological measurements, and subsequently generate nuisance datasets that only contain non-neural fluctuations. Using these datasets, we provide a comprehensive examination of the regional variability of the impact of the considered nuisance processes on the BOLD signal, as well as an investigation of the group consistency and inter-individual differences of their characteristic signatures on FC. We further evaluate several fMRI preprocessing strategies to assess the extent to which different techniques remove the physiological and motion FC signatures from the fMRI data. Finally, we investigate the potential effect of physiological processes and head motion on individual discriminability in the context of connectome fingerprinting.

Using the proposed approach, we show that SLFOs and head motion have a larger impact on FC measures compared to breathing motion and cardiac pulsatility, and we highlight the functional connections that are more prone to exhibiting biases. Furthermore, our findings suggest that the recurrent whole-brain connectivity patterns observed in time-varying FC can be partly attributed to SLFOs and head motion. Finally, we show that connectome fingerprinting accuracies are higher when non-neural confounds are reduced, suggesting a neural component underpinning the individual nature of FC patterns.

The codes that were employed to carry out the analyses described in the present study are publicly available and can be found on github.com/axifra/Nuisance_signatures_FC. (copy archived at swh:1:rev:52781e743d4b4eb491b9330210dac52dcd46fd10), *Xifra-Porxas, 2021a*.

## Results

### Contributions of nuisance processes to the BOLD signal

We examined regional differences in the influence of physiological processes and head motion to the BOLD signal. The physiological processes evaluated here were breathing motion, cardiac pulsatility, and SLFOs associated with changes in heart rate and breathing patterns (see Materials and methods – Nuisance processes evaluated). Scans with LR and RL phase encoding were examined

separately as it has been suggested that breathing motion artifacts vary across scans with different phase encoding directions (*Raj et al., 2001*), and thus we aimed to examine whether other processes such as head motion demonstrate a similar dependence. The contributions of each nuisance process on BOLD signal fluctuations were quantified as the correlation between the nuisance fluctuations of the process in question, modeled using externally recorded physiological measurements, and the BOLD fluctuations 'cleaned' of all other nuisance fluctuations (denoted as $r_{nuis}$ in the Materials and methods section – Isolation of nuisance fluctuations from fMRI data, see also Figure 7). We computed these contributions for each scan and then tested for the presence of consistent patterns across scans with the same phase encoding direction (significance testing using inter-subject surrogates, two-sample t-test, $p<0.05$, Bonferroni corrected).

The results showed distinct regional patterns for each of the nuisance processes. SLFOs mostly affected sensory regions, including the visual and somatosensory cortices (particularly of the face) (*Figure 1A*). Phase encoding was not found to modulate the magnitude of the SLFOs contributions on the BOLD signal (*Figure 1A*). Head motion exhibited the largest effect in the somatosensory and visual cortices (*Figure 1B*). Intriguingly, the effect in the visual cortex was highest in the right hemisphere for LR phase encoding, but highest in the left hemisphere for RL phase encoding (*Figure 1B*). Breathing motion effects were more pronounced in prefrontal, parietal and temporal brain regions (*Figure 1C*). Further, breathing motion had a much larger impact on the left hemisphere when the phase encoding was LR, whereas the reverse pattern was observed for RL phase encoding (*Figure 1C*). Cardiac pulsatility was highest in regions such as the visual and auditory cortices, as well as the insular cortex (*Figure 1D*).

## Physiological and head motion signatures in static FC

To examine the effect of physiological fluctuations and head motion in static FC, we developed a framework that quantifies the extent to which functional connections are influenced by a nuisance process at the individual scan level. Briefly, synthetic datasets were generated for each scan based on the contributions of the examined nuisance processes within each ROI (see Materials and methods – Isolation of nuisance fluctuations from fMRI data). These datasets retained the variance explained by nuisance fluctuations and replaced the remaining variance (often considered as the 'neural' variance) with random signals generated through an autocorrelated process. This framework allowed us to compute FC matrices that illustrate the whole-brain connectome profiles arising from the nuisance processes of interest (see Materials and methods – Estimation of static and time-varying functional connectivity).

The group-averaged static FC matrices across all 1568 scans revealed consistent whole-brain connectome patterns for SLFOs, head motion and breathing motion (*Figure 2A–C*). SLFO-based connectivity profiles exhibited strong positive correlations for all edges of the FC matrix, particularly for edges within the visual network, as well as between the visual network and the rest of the brain (*Figure 2A*). Head motion mainly influenced functional connections within the visual and sensorimotor networks, as well as edges within the DMN (*Figure 2B*). Note that even though areas in both the visual and sensorimotor networks were influenced by motion artifacts (*Figure 1B*), we did not observe strong correlations between the two aforementioned networks. This is not entirely surprising, as two brain areas may be associated with a different linear combination of head motion nuisance regressors and, thus, the correlation between the region-specific motion-induced fluctuations can in principle be around zero. Breathing motion exhibited an intriguing chess-like pattern, with both positive and negative correlations (*Figure 2C*, lower triangular matrix). Based on this observation, we subsequently reordered the ROIs with respect to their hemisphere, which revealed that positive correlations were mostly confined between ROIs of the same hemisphere, whereas correlations between hemispheres were close to zero or even negative (*Figure 2C*, upper triangular matrix). Even when scans with LR and RL phase encoding were averaged separately, both hemispheres exhibited increased within-hemisphere connectivity (*Figure 2—figure supplement 1C*). Nonetheless, the connectome profile of breathing motion exhibited clear differences between phase encoding directions, whereas all other nuisance processes did not exhibit perceivable differences (*Figure 2—figure supplement 1*). Finally, cardiac pulsatility did not exhibit a characteristic spatial pattern and the group-averaged correlation values were low, suggesting that it does not affect static FC in a systematic manner across subjects (*Figure 2D*).

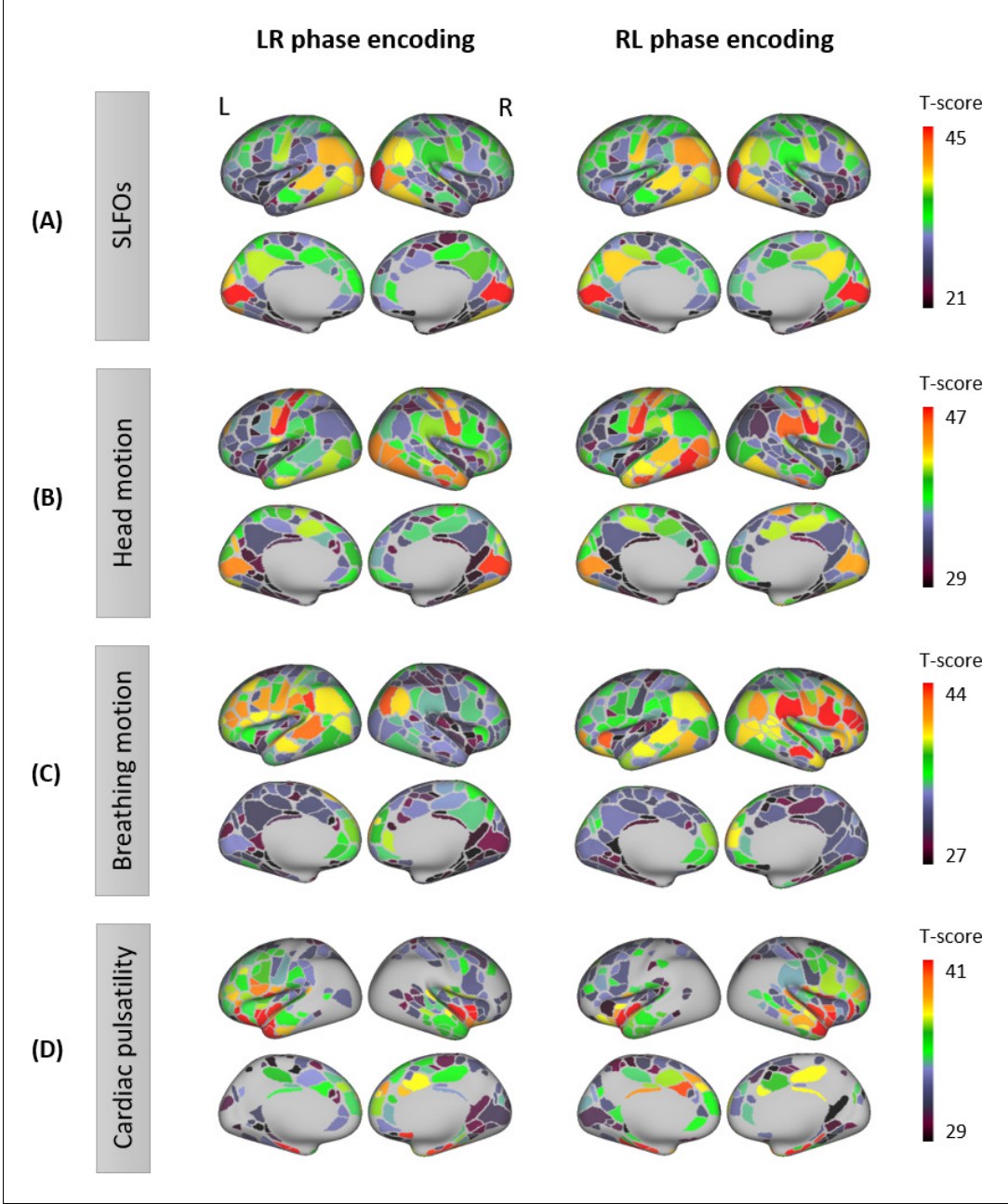

**Figure 1.** Contributions of nuisance processes to the resting-state BOLD signal. T-score maps of the correlation between each nuisance process and BOLD fMRI fluctuations (raw data) for (**A**) SLFOs, (**B**) head motion, (**C**) breathing motion, and (**D**) cardiac pulsatility, computed within each parcel of the Gordon atlas (nonparametric permutation test, p<0.05, FDR corrected). The tests were performed for each phase encoding separately. The physiological fluctuations were obtained from simultaneous external recordings. These results illustrate the cortical regions most affected by each nuisance process.

## Capability of preprocessing strategies to remove the nuisance signatures on static FC

To examine the capability of various preprocessing strategies to reduce the effects introduced by physiological processes and head motion on static FC, we computed for each scan the similarity of the connectome profile that arises from a nuisance process with the connectome profile calculated from preprocessed fMRI data (considered as the 'neural' profile). This similarity reflects the extent to which the 'neural' connectome profile extracted after a specific denoising strategy is confounded by

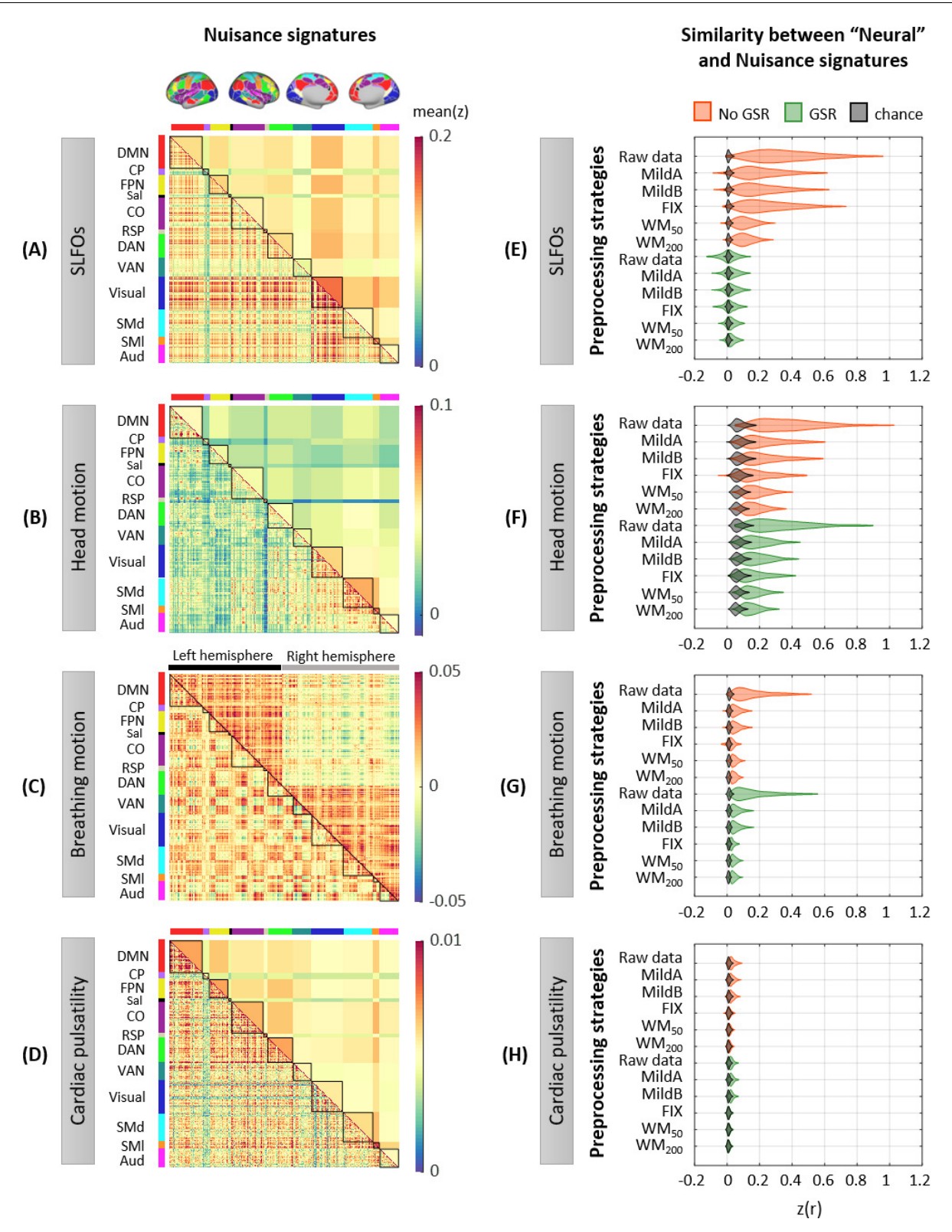

**Figure 2.** Whole-brain connectome patterns induced by nuisance processes and effect of preprocessing strategies. (A–D) Group averaged nuisance FC matrices across all 1568 scans for (A) SLFOs, (B) head motion, (C) breathing motion, and (D) cardiac pulsatility. The lower triangular matrices show the FC value for each network pair. The upper triangular matrices show FC values from the lower triangular averaged within network pairs (A, B, D panels), or the FC value for each network pair reordered according to left/right hemisphere (C panel). These results demonstrate that nuisance fluctuations

*Figure 2 continued on next page*

*Figure 2 continued*

induce heterogeneous whole-brain connectivity profiles which, if unaccounted for, can result in biased estimates functional connectivity. (E–H) Distribution of Pearson correlation coefficients across all 1568 scans between the 'neural' FC matrix for different preprocessing strategies and nuisance FC matrices associated to (E) SLFOs, (F) head motion, (G) breathing motion, and (H) cardiac pulsatility. Correlation values were Fisher z transformed. SLFOs, head motion and breathing motion were found to confound the FC matrices more severely (E–G). GSR effectively removed the effects of SLFOs, while more aggressive preprocessing pipelines mitigated the effects of head motion, breathing motion and cardiac pulsatility.

The online version of this article includes the following figure supplement(s) for figure 2:

**Figure supplement 1.** Whole-brain connectome patterns induced by nuisance processes, separately for scans with LR and RL phase encoding.

**Figure supplement 2.** Effectiveness of preprocessing strategies in reducing the whole-brain connectivity profiles effects of SLFOs for two different global signal calculation methods.

**Figure supplement 3.** Effectiveness of preprocessing strategies in reducing the whole-brain connectivity effect of each nuisance process, with and without including model-based regressors.

**Figure supplement 4.** Investigating the negative correlations between the 'neural' and SLFOs signatures after GSR.

**Figure supplement 5.** Whole-brain connectome patterns induced by nuisance processes and effect of preprocessing strategies, using the Seitzman atlas.

physiological and head motion artifacts. A distribution of the similarity values across scans, in this case Pearson's correlation coefficients, is shown in *Figure 2E–H* for each preprocessing strategy and nuisance process. We found that SLFOs, head motion and breathing motion had the strongest influence on static FC, based on the similarity of their connectome profiles with the 'neural' connectome profiles from the raw data (*Figure 2E–H*).

The signature induced by SLFOs remained after the MildA, MildB, and FIX pipelines were applied, but was greatly reduced by the $WM_{50}$ and $WM_{200}$ strategies (*Figure 2E*). The observation that FIX, which is a rather aggressive preprocessing strategy, was unable to remove most of the SLFOs is consistent with recent studies showing that global artifactual fluctuations are still prominent after FIX denoising (*Burgess et al., 2016*; *Glasser et al., 2018*; *Kassinopoulos and Mitsis, 2019*; *Power et al., 2018*; *Power et al., 2017*). Notably, GSR seemed to be an effective technique for removing the physiological signature from SLFOs on static FC, albeit for some scans it appeared to introduce a negative correlation between the SLFOs and 'neural' FC matrices (*Figure 2E*). This effect was greater when the global signal was computed across the whole brain in volumetric space, compared to across vertices in surface space (*Figure 2—figure supplement 2A*). The effect of the signature related to head motion was reduced with more aggressive preprocessing strategies, but none of the examined approaches completely eradicated the head motion effects in static FC (*Figure 2F*). GSR slightly reduced the similarity between the head motion and 'neural' connectome profiles. The signature induced by breathing motion was greatly reduced by all preprocessing strategies, and particularly by FIX denoising, which yielded almost chance level (*Figure 2G*). Still, none of the preprocessing strategies entirely eliminated the breathing motion signature in static FC. The confounds introduced by cardiac pulsatility were overall small and effectively removed by the FIX, $WM_{50}$, and $WM_{200}$ strategies (*Figure 2H*). GSR did not have any effect on the removal of breathing motion and cardiac pulsatility connectome profiles.

Finally, we evaluated the addition of model-based nuisance regressors to the preprocessing strategies. Specifically, we added the physiological regressors used to model SLFOs (*Kassinopoulos and Mitsis, 2019*), cardiac pulsatility and breathing motion (*Glover et al., 2000*). We found that including the regressor that models SLFOs reduces their effect on static FC for all preprocessing strategies apart from $WM_{50}$ and $WM_{200}$, but, in contrast to GSR, the similarity remains well above chance levels (*Figure 2—figure supplement 3A*). Including the RETROICOR regressors related to breathing motion considerably reduced the breathing motion signature in the raw data and when only using GSR as a preprocessing method; however, none of the preprocessing strategies benefited from including these regressors (*Figure 2—figure supplement 3C*). On the contrary, including the RETROICOR regressors related to cardiac pulsatility completely removed the effect of the latter for the raw data and the MildA and MildB strategies (*Figure 2—figure supplement 3D*), suggesting that conservative preprocessing strategies greatly benefit by adding the model-based regressors for cardiac pulsatility.

## Connectome-based identification of individuals

We next investigated the extent to which FC matrices associated to physiological processes and head motion can identify an individual subject, and whether the accuracy of connectome-based fingerprinting is inflated by the examined nuisance processes (see Materials and methods – Connectome-based identification of individual subjects).

We initially considered all the edges of the FC matrices for subject identification (Gordon atlas: 40,755 edges). Accuracy was above chance for all database-target combinations for the nuisance processes, with rates up to 40% (*Figure 3A*). Breathing motion exhibited an intriguing bimodal distribution: database-target pairs that had the same phase encoding yielded much higher identification rates than database-target pairs with different phase encodings, even if the latter were acquired on the same day. This effect was also observed, although to a lesser extent, for cardiac pulsatility and head motion.

Identification accuracy was much higher for the 'neural' datasets compared to the nuisance datasets, with rates ranging from 52% to 99% (*Figure 3B*). The MildA, MildB, and FIX techniques considerably improved the accuracy compared to the raw data, and the $WM_{50}$ and $WM_{200}$ techniques significantly outperformed all other preprocessing strategies. GSR considerably improved identification accuracy for the MildA, MildB and FIX strategies. Furthermore, we observed that for the raw data, database-target pairs from different days with the same phase encoding showed identification rates as high as the ones from same day but different phase encoding. In contrast, for all other preprocessing strategies the database-target pairs from the same day were always higher.

We subsequently tested identification accuracy on the basis of within and between edges of specific functional networks to examine whether certain functional connections had a more pronounced contribution to individual subject discriminability. Results for the nuisance datasets are shown in *Figure 4A*, where it can be seen that nuisance processes yielded a markedly lower identification accuracy when using specific edges compared to using all edges (*Figure 3A*). Furthermore, functional connections between networks seemed to contribute more to the subject discriminability of SLFOs compared to connections within brain networks (p<0.001, Wilcoxon rank-sum).

Regarding the 'neural' datasets, we focused on the most aggressive strategies, namely FIX and $WM_{200}$. Networks of 'top-down' control (FPN, CO, DAN, VAN), as well as the DMN, yielded higher identification accuracy compared to sensorimotor processing networks (Visual, SMd, Aud) for all preprocessing strategies (*Figure 4B*). These results indicate that FC patterns in higher order association cortices ('top-down' control networks) tend to be distinctive for each individual, whereas primary sensory and motor regions (processing networks) tend to exhibit similar patterns across individuals, consistent with previous studies (*Finn et al., 2015*; *Gratton et al., 2018*; *Horien et al., 2019*). We then tested for differences in identification accuracy between the raw data vs. FIX and $WM_{200}$ data (*Figure 4C*, p<0.05, Bonferroni corrected, Wilcoxon rank-sum). FIX denoising significantly increased the subject discriminability of connections within and between several top-down control networks, but significantly decreased the subject discriminability of connections between the FPN and SMd, as well as the FPN and Aud networks. Conversely, $WM_{200}$ denoising significantly increased the subject discriminability of connections within and between all top-down control networks, connections within the Visual and SMd networks, and connections of the DAN with the Visual and SMd network.

## Physiological and head motion signatures in time-varying FC and the effect of preprocessing strategies

To examine the effect of physiological processes and head motion on time-varying FC estimates, we computed functional connectivity dynamics (FCD) matrices (*Hansen et al., 2015*) using the generated nuisance and 'neural' datasets from each scan, whereby each FCD matrix captures the temporal evolution of FC patterns within a scan. We subsequently computed the similarity of the nuisance and 'neural' FCD matrices at the individual level to examine the capability of various preprocessing strategies to reduce the confounds introduced by physiological processes and head motion on time-varying FC. A distribution of the similarity values, in this case Pearson's correlation coefficients, is shown in *Figure 5* for each preprocessing strategy and nuisance process. We observed that the temporal evolution of FC patterns from SLFOs and head motion were similar to the ones observed in the raw data. An illustration of this similarity is shown in *Figure 6* for six subjects. On the other hand, the distribution of similarity values for breathing motion and cardiac pulsatility FCD matrices was around

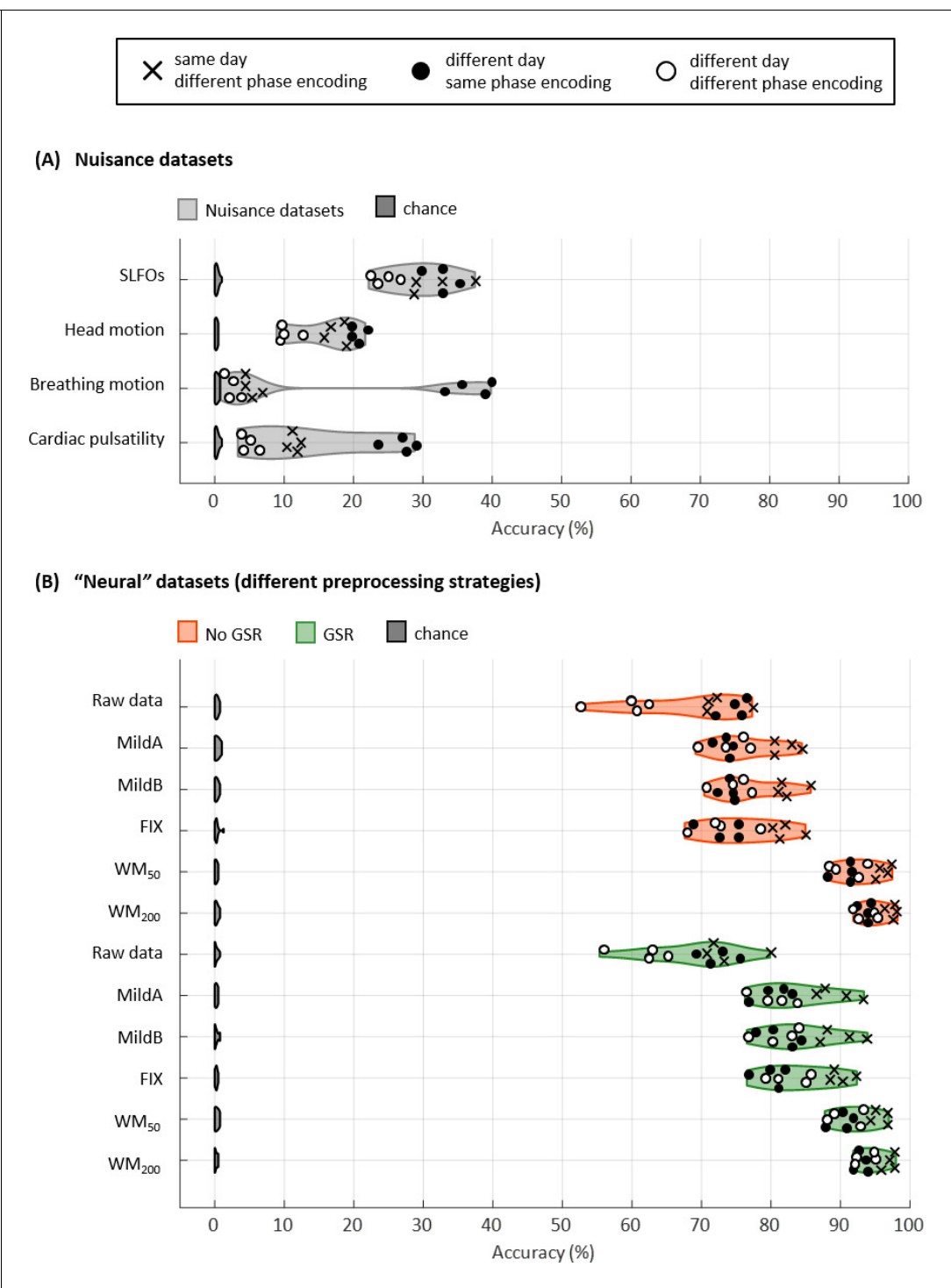

**Figure 3.** Connectome fingerprinting results. (**A**) Fingerprinting accuracy obtained using the static FC matrices from the generated nuisance datasets whereby non-neural fluctuations were isolated from the BOLD data. Above-chance level accuracy values were obtained for all nuisance processes, suggesting some degree of subject specificity in whole-brain connectivity profiles arises from nuisance fluctuations. (**B**) Fingerprinting accuracy obtained using the static FC matrices generated from each of the preprocessing strategies evaluated. The pairs of resting-state scans are indicated with different symbols, depending on whether they belong to the same or different day session, as well as whether they have the same phase encoding. Higher fingerprinting accuracy values were observed for white matter denoising approaches ($WM_{50}$, $WM_{200}$) compared to milder pipelines and FIX denoising. Both mild and more aggressive pipelines yielded higher subject discriminability for pairs of scans acquired on the same day. GSR increased the fingerprinting accuracy of milder strategies and FIX denoising.

The online version of this article includes the following figure supplement(s) for figure 3:

*Figure 3 continued on next page*

*Figure 3 continued*
**Figure supplement 1.** Schematic of the methodology for connectome fingerprinting.
**Figure supplement 2.** Connectome fingerprinting results for the Seitzman atlas.

zero, indicating the absence of systematic effects on 'neural' FCD matrices (*Figure 5*). Including model-based regressors in the preprocessing pipelines led to a small decrease in the similarity between neural and SLFOs FCD matrices, particularly for mild pipelines and FIX without GSR, even

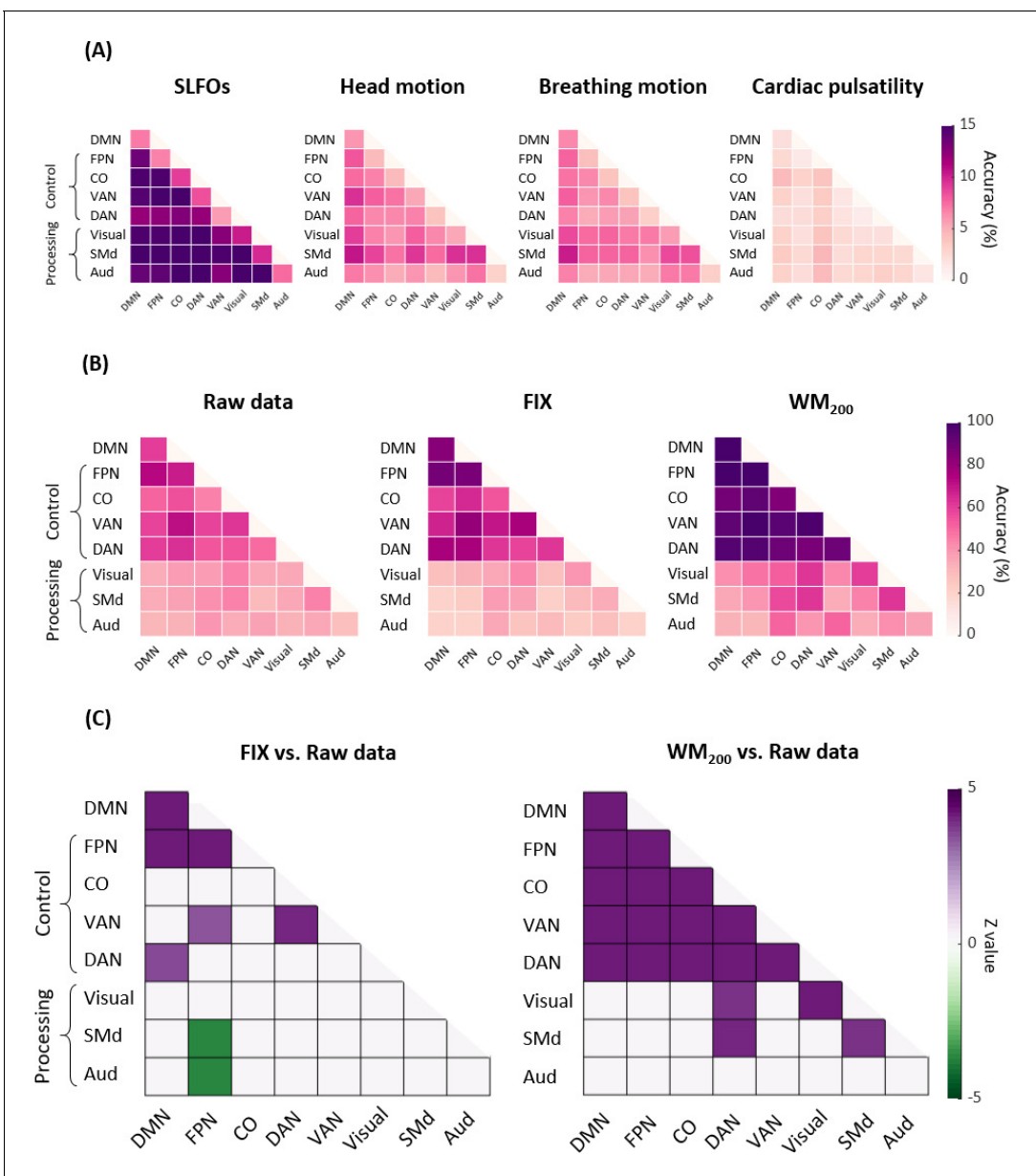

**Figure 4.** Connectome fingerprinting results using edges within and between networks. (A) Fingerprinting accuracy for SLFOs, head motion, breathing motion and cardiac pulsatility, averaged across all database-target pairs. (B) Fingerprinting accuracy for the raw data, FIX and WM$_{200}$ pipelines, averaged across all database-target pairs. (C) Significant differences in fingerprinting accuracy obtained when using the FIX and WM$_{200}$ pipelines as compared to the raw data (p<0.05, Bonferroni corrected, Wilcoxon rank-sum). Connectivity profiles within and between top-down control networks (FPN, CO, VAN, DAN) and DMN yielded higher identification accuracy compared to connectivity profiles within and between sensorimotor processing networks (Visual, SMd, Aud).

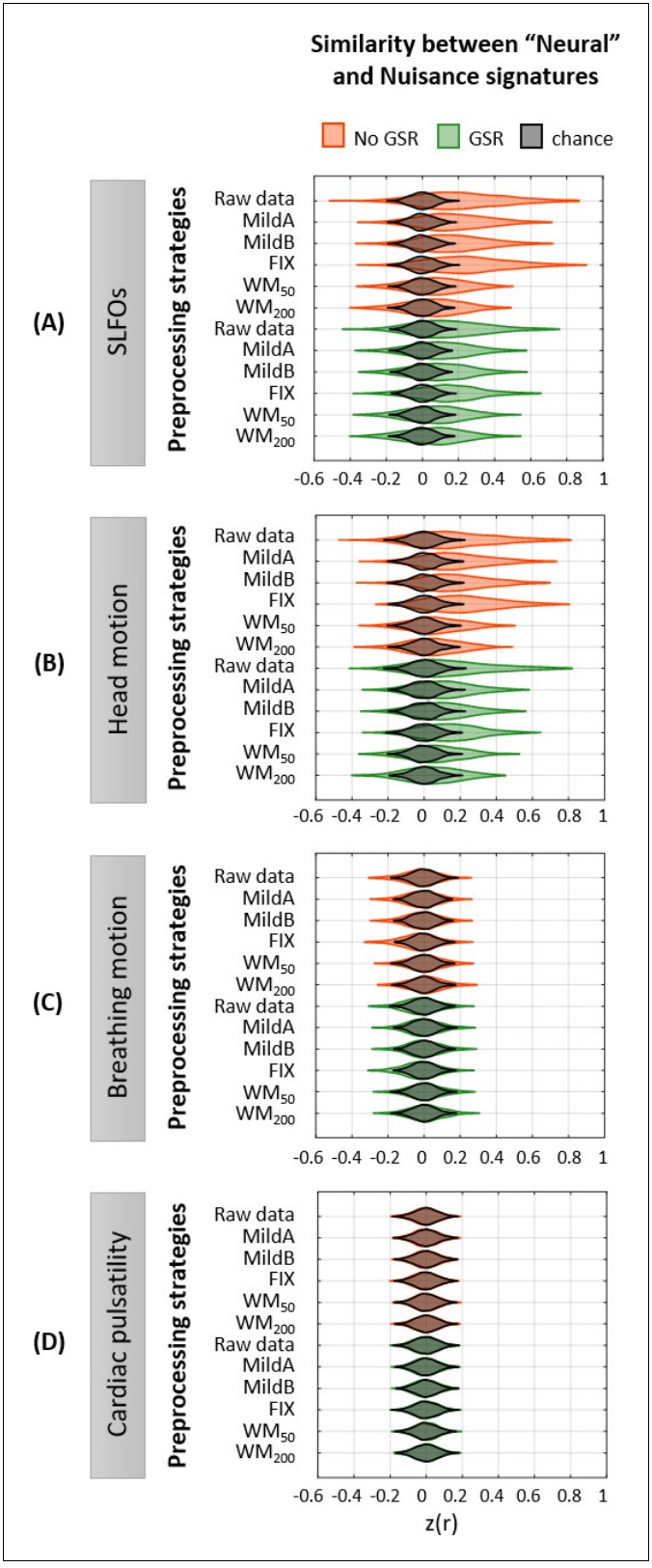

**Figure 5.** Effectiveness of preprocessing strategies in reducing functional connectivity dynamics (FCD) profiles induced by physiological and motion processes. Distribution of Pearson correlation coefficients across all 1568 scans between the 'neural' functional connectivity dynamics (FCD) matrix after each preprocessing pipeline and nuisance FCD matrices associated to (A) SLFOs, (B) head motion, (C) breathing motion, and (D) cardiac pulsatility. *Figure 5 continued on next page*

*Figure 5 continued*

Correlation values were Fisher z transformed. Results shown in the top row of each subpanel (raw data) suggest that SLFOs and head motion most severely confound the FC matrices, whereas breathing motion and cardiac pulsatility do not induce artifactual dynamics. None of the examined strategies completely eliminated these effects.

The online version of this article includes the following figure supplement(s) for figure 5:

**Figure supplement 1.** Effectiveness of preprocessing strategies in reducing functional connectivity dynamics (FCD) profiles induced by physiological and motion processes, with and without including model-based regressors.

**Figure supplement 2.** Effectiveness of preprocessing strategies in reducing functional connectivity dynamics (FCD) profiles induced by physiological and motion processes, using the Seitzman atlas.

though these decreased similarity values were still above chance levels (*Figure 5—figure supplement 1A*).

None of the preprocessing pipelines was able to vanish the effects of SLFOs and head motion. However, these effects were considerably reduced by the WM$_{50}$ and WM$_{200}$ strategies (*Figure 5A–B*, *Figure 6*). FIX denoising was the least successful strategy in terms of reducing the SLFOs' signature (*Figure 5A*, *Figure 6*), similarly to static FC (*Figure 2E*), and only achieved the same levels of performance as other strategies after GSR. However, even after GSR none of the strategies reached chance levels (*Figure 5A*), in contrast with the static FC results (*Figure 2E*). GSR led also to a slight reduction in the similarity between the head motion and 'neural' FCD matrices (*Figure 5B*), as in the case of static FC (*Figure 2F*).

## Discussion

In this work, we characterized the effects of physiological processes and head motion on static and time-varying estimates of functional connectivity measured with BOLD fMRI. While the BOLD signal is considered a proxy of neural activity via changes in local blood oxygenation, physiological processes and motion artifacts can also induce variations in the BOLD signal, which can in turn lead to confounds in estimates of functional connectivity. Here, we developed an innovative framework to characterize the spatial signature of head motion and physiological processes (cardiac and breathing activity) on estimates of functional connectivity. Our results demonstrated that functional connectivity measures can be influenced by non-neural processes. Specifically, we identified stereotyped whole-brain functional connectivity profiles for SLFOs, head motion and breathing motion (*Figure 2A–C*), suggesting that these processes introduce a systematic bias in estimates of functional connectivity if they are not properly accounted for. Furthermore, we provided evidence that recurring patterns in time-varying FC can be attributed, to some extent, to SLFOs and head motion (*Figure 5*, *Figure 6*). We also assessed the performance of several state-of-the-art preprocessing strategies in mitigating the effects of nuisance processes, and showed that more aggressive preprocessing strategies such as FIX (*Salimi-Khorshidi et al., 2014*) and WM denoising (*Kassinopoulos and Mitsis, 2021a*) combined with GSR were the most effective with regard to removing the effects of non-neural processes for both static and time-varying FC analyses (*Figure 2E–H*, *Figure 5*, *Figure 6*). Finally, we evaluated the potential subject specificity of the connectivity profiles associated with physiological and motion confounds, along with their role as hypothetical contributors to connectome fingerprinting accuracy. Interestingly, we found that these non-neural functional connectivity patterns are to some extent subject specific (*Figure 3A*); however, fMRI data corrected for these confounds increased identification accuracy in connectome fingerprinting (*Figure 3B*), suggesting that the inter-individual differences in FC that facilitate subject identification are strongly neural and do not largely stem from physiological processes or head motion.

### Spatially heterogeneous contributions of nuisance processes to the BOLD signal

It is well established that head and breathing motion affect areas at the edges of the brain (*Jo et al., 2010*; *Patriat et al., 2015*; *Satterthwaite et al., 2013*), whereas cardiac pulsatility affects areas near the large cerebral arteries just above the neck (*Glover et al., 2000*; *Kassinopoulos and Mitsis,*

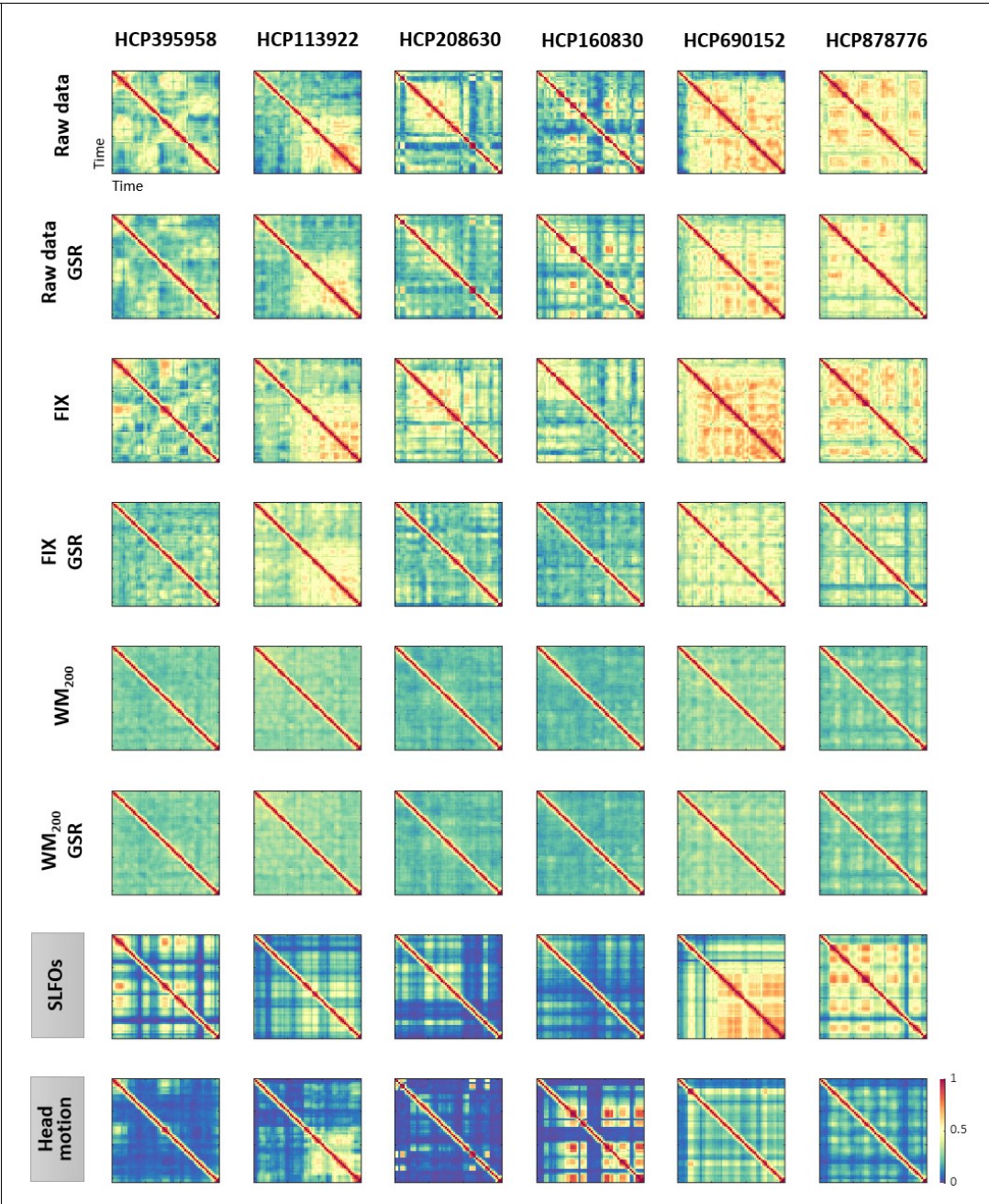

**Figure 6.** Functional connectivity dynamics (FCD) profiles associated with SLFOs and head motion resemble patterns commonly attributed to neural processes. Illustrative examples of FCD matrices from specific HCP subjects as obtained from the fMRI data for several pre-processing pipelines (rows 1–6), as well as from SLFOs and head motion (rows 7 and 8, respectively). All the examples are from the HCP scan Rest1_LR. These examples show a clear resemblance between FCD matrices computed from the 'neural' datasets and the nuisance processes (SLFOs and head motion). Note that the size of FCD matrices is W×W, where W is the number of sliding windows within a scan.

The online version of this article includes the following figure supplement(s) for figure 6:

**Figure supplement 1.** Examples of functional connectivity dynamics (FCD) profiles.

2021b). These observations are based on studies that typically examine the brain regions affected by the aforementioned sources of noise on a voxel-wise basis. However, at the voxel level we cannot easily assess whether the average fMRI signal from atlas-based ROIs includes significant contributions from these nuisance processes. In principle, it could be the case that the dynamics of artifacts associated with a specific nuisance process demonstrate significant variability across voxels, and as a

result their effects cancel out when averaging voxels within an ROI. In the present study, we assessed the impact and regional variation of these nuisance processes in the Gordon parcellation (*Gordon et al., 2016*), a widely used atlas in the literature.

SLFOs related to changes in heart rate and breathing patterns were found to affect mostly sensory regions including the visual and somatosensory cortices (particularly of the face) (*Figure 1A*), which correspond to regions with a high density of veins (*Bernier et al., 2018*; *Huck et al., 2019*). The spatial pattern of SLFOs is very similar to statistical maps reported in prior works, which have highlighted brain regions highly correlated with the global signal (*Billings and Keilholz, 2018*; *Glasser et al., 2016*; *Li et al., 2019a*; *Power et al., 2017*; *Tong et al., 2013*; *Zhang et al., 2020*). This is not surprising, since the global signal is strongly driven by fluctuations in heart rate and breathing patterns (*Birn et al., 2006*; *Chang and Glover, 2009a*; *Falahpour et al., 2013*; *Kassinopoulos and Mitsis, 2019*; *Shmueli et al., 2007*). Moreover, we show evidence that SLFOs and cardiac pulsatility do not affect the same brain regions, consistent with (*Chen et al., 2019*; *Kassinopoulos and Mitsis, 2019*; *Tong and Frederick, 2014*). Specifically, cardiac pulsatility was more dominant in regions such as the insular and auditory cortices, which align with cortical branches of the middle cerebral artery (*Figure 1D*) and are the regions with highest arterial density (*Bernier et al., 2018*).

Regarding head motion, previous studies found that its effect was more pronounced in prefrontal, sensorimotor, and visual brain regions (*Satterthwaite et al., 2013*; *Yan et al., 2013*). However, these studies did not remove breathing artifacts from the realignment parameters, which are present even in single-band datasets (*Gratton et al., 2020*), and thus were unable to disentangle whether a specific type of motion affected particular brain regions. In the present work, we regressed out breathing motion from the realignment parameters, and observed that sensorimotor and visual areas were strongly affected by head motion (*Figure 1B*), whereas breathing motion artifacts were more pronounced in the prefrontal cortex and brain regions in the parietal and temporal cortices (*Figure 1C*). Furthermore, Yan et al. showed that framewise displacement was positively correlated with sensory regions and negatively correlated with prefrontal regions. Collectively, these findings suggest that most regions exhibit an increase in the BOLD signal due to head and breathing motion, whereas the prefrontal cortex may exhibit a decrease in the BOLD signal likely due to breathing-related chest movements.

## Physiological and head motion signatures in static FC

Head motion is considered the biggest source of confound for FC fMRI studies and there is a significant effort from the neuroimaging community toward developing and evaluating preprocessing strategies that mitigate its effects (*Ciric et al., 2017*; *Parkes et al., 2018*; *Power et al., 2015*). On the other hand, while it has been shown that SLFOs affect the default-mode network (*Birn et al., 2014*; *Birn et al., 2008a*; *Chang and Glover, 2009a*), and high frequency cardiac and breathing artifacts influence the BOLD signal (*Glover et al., 2000*; *Power, 2019*), a systematic investigation of the effects of physiological processes in the context of whole-brain FC is lacking in the literature. In the present study, we evaluated collectively the impact of the aforementioned sources of noise on whole-brain fMRI resting-state FC.

Our results revealed that all four nuisance datasets exhibited mainly positive correlations between ROIs (*Figure 2A–D*), suggesting that the presence of nuisance fluctuations in a conventional fMRI dataset typically leads to a shift of correlation values toward more positive numbers. In other words, in the case of an fMRI dataset that has not been corrected for nuisance fluctuations, two ROIs for which neural-related fluctuations are negatively correlated could be found to be positively correlated due to the presence of similar nuisance fluctuations in the ROIs. Furthermore, we observed that SLFOs and head motion confounded FC to a larger degree compared to breathing motion and cardiac pulsatility (*Figure 2E–H*).

Our results suggest that SLFOs due to spontaneous changes in heart rate or breathing patterns inflate connectivity (toward more positive values) across the whole brain but particularly for edges within the visual network, as well as edges between the visual and the rest of the networks (*Figure 2A*). It is well known that the visual cortex is characterized by the highest venous density (*Bernier et al., 2018*), possibly due to its functional importance (*Collins et al., 2010*). In addition, it has been shown that brain regions with higher vascular density exhibit larger amplitude of spontaneous BOLD fluctuations (*Vigneau-Roy et al., 2014*). Therefore, it is likely that the structure of the

SLFOs' connectome profile may largely reflect the underlying vascular architecture. The effect of SLFOs on static FC was considerably reduced after WM denoising, while additionally performing GSR almost removed this effect (*Figure 4E*). Notably, FIX denoising without GSR was unable to remove the confounds introduced by SLFOs, which is consistent with recent studies showing that global artifactual fluctuations are still prominent after FIX denoising (*Burgess et al., 2016*; *Glasser et al., 2018*; *Kassinopoulos and Mitsis, 2019*; *Power et al., 2018*; *Power et al., 2017*).

Head motion was found to influence the connectivity within the visual and sensorimotor networks (*Figure 2B*), in line with previous studies (*Power et al., 2012*; *Satterthwaite et al., 2012*; *Van Dijk et al., 2012*). Our results showed that only regressing out the realignment parameters and average WM/CSF signals (with or without expansion terms) is not sufficient to remove the effects of head motion (*Figure 2F*, MildA and MildB pipelines), which is consistent with findings in *Parkes et al., 2018*. Among all preprocessing strategies, WM denoising yielded the largest reduction of motion effects (*Figure 2F*). The two pipelines WM$_{50}$ and WM$_{200}$ refer to the removal of 50 and 200 white matter regressors from the data (i.e. principal components obtained from the white matter compartment). In our previous study, we showed that while both pipelines yielded high large-scale network identifiability compared to other pipelines, the more aggressive WM$_{200}$ resulted in a larger reduction of motion artifacts compared to WM$_{50}$ (*Kassinopoulos and Mitsis, 2021a*). The results of the current study also show a stronger reduction of head motion effects for the former compared to the latter (*Figure 2F*), which may explain the higher accuracy in connectome fingerprinting observed for the pipeline WM$_{200}$ compared to WM$_{50}$ (*Figure 3B*).

A natural concern regarding the head motion connectome profile is that it may reflect motor-related activity (*Yan et al., 2013*). Even though motor-related neural activity would be expected to lag the instantaneous motion traces due to the sluggishness of the hemodynamic response, we cannot exclude the scenario that the head motion connectome profile reflects the neural correlates of the executed movements and eye adjustments to fixate on the cross. Nonetheless, even if preprocessing strategies remove neural activity associated with spontaneous head movements, this source of neural activity is typically of no interest in resting-state fMRI studies.

Furthermore, we provide evidence that head and breathing motion do not affect functional connectivity in the same manner. Specifically, breathing motion was found to inflate within-hemisphere connectivity (*Figure 2C*). This bias seems to arise as a result of factitious motion rather than real motion of the head, since it is related to the LR/RL phase encoding direction (see section 4.4 for more details). All preprocessing strategies yielded a substantial reduction of artifacts related to breathing motion, with FIX denoising being the most effective (*Figure 2G*).

In our dataset, cardiac pulsatility did not seem to have a large effect on FC, neither in cortical nor subcortical regions (*Figure 2D*, *Figure 2—figure supplement 5D*), and its effect was entirely removed with more aggressive pipelines such as FIX and WM denoising (*Figure 2H*), as well as with model-based techniques (*Figure 2—figure supplement 3D*). However, it has been recently reported that the 3T HCP dataset has poor temporal signal-to-noise ratio in the subcortex (*Ji et al., 2019*; *Seitzman et al., 2020*). Therefore, it is possible that we may have underestimated the effect of cardiac pulsatility in functional connections involving subcortical regions.

It is important to note that our proposed methodology assumes that the stereotyped nuisance connectome profiles do not resemble the true neural connectome profiles. However, in principle, nuisance fluctuations could give rise to similar spatial patterns as neurally-driven fluctuations. A recent study by *Bright et al., 2020* provided evidence that physiological fluctuations (end-tidal $CO_2$) give rise to networks that spatially resemble neurally driven networks linked to working memory and visual stimuli. The authors suggested that this phenomenon may be due to the vasculature adapting to the neural network architecture, as vascular and neuronal growth processes evolve concurrently during development (*Quaegebeur et al., 2011*). These findings suggest a possible caveat of our methodology when assessing pre-processing strategies, as pipelines that yield the lowest similarity between nuisance and 'neural' FC matrices might also remove some signal of interest. Nonetheless, the pre-processing strategies that were found in this study to reduce the nuisance effects the most (i.e. FIX and WM denoising combined with GSR) have been shown to demonstrate the highest improvement in large-scale network identifiability in an earlier study (*Kassinopoulos and Mitsis, 2021a*). In addition, these pipelines were found to exhibit the highest accuracy in connectome fingerprinting (*Figure 3B*). These results suggest that they are able to adequately remove the effects of nuisance processes while also preserving the signal of interest.

## Physiological and head motion signatures in time-varying FC

The investigation of neural dynamics using resting-state fMRI is a promising avenue of research that has gained increasing attention lately (*Hutchison et al., 2013*; *Lurie et al., 2020*). Yet, there is skepticism regarding its validity and underlying origins. For instance, variations in FC over shorter timescales (i.e. minutes) could largely be explained by sampling error, acquisition artifacts and subject arousal (*Hindriks et al., 2016*; *Laumann et al., 2017*; *Savva et al., 2020*), as well as head motion and physiological processes (*Nalci et al., 2019*; *Nikolaou et al., 2016*).

In the present study, we sought to evaluate whether non-neural fluctuations could partly explain the recurrent connectivity patterns observed in fMRI studies. To this end, we computed time-resolved FC dynamics (*Hansen et al., 2015*) for all four nuisance datasets, and assessed their similarity with time-resolved FC dynamics obtained from fMRI data preprocessed employing widely used denoising strategies ('neural' datasets). The FC dynamics of head motion and SLFOs datasets were markedly similar to the FC dynamics observed in preprocessed fMRI data (*Figure 5A–B*), albeit the similarity was smaller compared to static FC (*Figure 2E–F*). When observing the time-resolved FC matrices (*Figure 6*), it becomes apparent that a large component of variability in FC patterns is due to non-neural processes, and that these patterns remain after implementing popular preprocessing pipelines such as MildA, MildB and FIX. These results are aligned with the observation that even after regressing out nuisance processes from the BOLD signal, correlations between time-varying FC measures and nuisance fluctuations remain (*Nalci et al., 2019*). WM denoising was found to be the most efficient strategy in terms of mitigating the influence of nuisance processes on time-varying FC (*Figure 5A–B*).

After WM denoising, variability in FC patterns was greatly diminished (*Figure 6*), even when those patterns could not be directly associated with any nuisance process (*Figure 6—figure supplement 1*). These results can be interpreted in two ways that are not mutually exclusive: (1) A significant fraction of the variability in FC patterns is a result of non-neural confounds and WM denoising is able to remove most of these confounds. This is supported by the fact that many other nuisance processes, which we did not examine here (e.g. arterial blood pressure, $CO_2$ concentration, scanner instabilities), can influence the BOLD signal and time-varying FC patterns (*Nikolaou et al., 2016*; *Whittaker et al., 2019*; *Wise et al., 2004*). (2) WM denoising removes a considerable fraction of variance of neural origin. Future work with concurrent direct measurements of neuronal activity (e.g. electroencephalography, calcium imaging) and additional physiological recordings would be instrumental for resolving to which extent time-varying FC is the result of underlying neural dynamics.

While head motion and SLFOs were found to be strongly associated to recurrent connectivity patterns, breathing motion and cardiac pulsatility do not seem to be a main concern for time-varying FC studies (*Figure 5C–D*). Likely, the effects of breathing motion and cardiac pulsatility do not influence time-varying FC, because their effect on the BOLD signal does not change from window to window, possibly due to their quasi-periodic nature. In contrast, the levels of head motion vary across time windows, which can modulate time-varying FC patterns. Heart rate and breathing patterns can be relatively constant during some time periods, whereby SLFOs are not expected to influence the BOLD signal and, in turn, the FC measures across time windows. On the other hand, in other instances heart rate and breathing patterns may change considerably over time, whereby SLFOs are expected to influence the BOLD signal and thus modulate the FC measures across time windows. In other words, ROIs sensitive to head motion and SLFOs are likely to exhibit a time-varying signal-to-noise ratio depending on the presence of these sources of noise, which eventually leads to confounds in time-varying FC measures.

Importantly, neither data-driven nor model-based preprocessing strategies were able to completely remove these confounds (*Figure 5—figure supplement 1*). It was only recently that researchers have started to examine the performance of pre-processing pipelines in the context of time-varying FC (*Lydon-Staley et al., 2019*), albeit with a focus on motion effects, thus more work is needed to identify effective data cleaning strategies for resting-state time-varying FC studies.

## Global signal regression

The practice of removing the GS from fMRI data (i.e. GSR) has been adopted by many fMRI investigators as it has been linked to head motion artifacts and fluctuations in heart rate and breathing patterns (*Birn et al., 2006*; *Byrge and Kennedy, 2018*; *Chang and Glover, 2009a*; *Falahpour et al.,*

*2013*; *Kassinopoulos and Mitsis, 2019*; *Power et al., 2018*; *Power et al., 2014*; *Shmueli et al., 2007*). Further, GSR has been shown to increase the neuronal-hemodynamic correspondence of FC measures extracted from BOLD signals and electrophysiological high gamma recordings (*Keller et al., 2013*), as well as strengthen the association between FC and behavior (*Li et al., 2019b*). On the other hand, studies capitalizing on EEG-fMRI data have reported an association between the GS amplitude and vigilance measures (*Wong et al., 2016*; *Wong et al., 2013*) and individual differences in the global signal topography have been related to behavior and cognition (*Li et al., 2019a*). Thus, as there is evidence that GSR may remove neuronal-related activity in addition to nuisance-related fluctuations, GSR still remains a controversial pre-processing step (*Liu et al., 2017*; *Murphy et al., 2009*; *Murphy and Fox, 2017*).

Our results provide evidence that, in the context of static FC, GSR removes physiological fluctuations related to SLFOs and to a lesser extent head motion artifacts (*Figure 2E–F*). Note that GSR does not account for breathing motion artifacts (*Figure 2G*) but rather changes in breathing patterns and deep breaths, which are related to SLFOs and possibly the head motion component at ~0.12 Hz (*Power et al., 2019*). Furthermore, GSR improved connectome fingerprinting accuracy (*Figure 3B*), which suggests that by removing nuisance fluctuations due to SLFOs and head motion, GSR enhances the individual specificity of connectivity profiles. Overall, our results suggest that the strong reduction in the effects of SLFOs and head motion achieved by GSR outweighs the possible loss of neuronal-driven fluctuations when examining FC patterns. GSR is particularly important when using ICA-based noise correction techniques such as FIX and AROMA (*Pruim et al., 2015*; *Salimi-Khorshidi et al., 2014*), since ICA components related to SLFOs frequently exhibit similar spatial patterns and frequency profile to neural components and thus are classified as non-artifactual and remain in the data after denoising.

For some scans, GSR led to a negative similarity between the SLFOs and neural signatures (*Figure 2E*). Further investigation revealed that scans that exhibited a negative SLFOs-neural similarity after GSR were scans where a large fraction of the GS variance was attributed to SLFOs (*Figure 2—figure supplement 4A*). In contrast, scans with low correlation between GS and SLFOs, which are likely scans with fairly constant cardiac and breathing rhythms, were not affected by GSR. In addition, we observed that in the case of scans with negative SLFOs-neural similarity after GSR, the estimated SLFOs signature yielded relatively low contributions to the somatosensory and auditory network compared to other networks (*Figure 2—figure supplement 4B*, left), while these two networks exhibited high correlation values for the neural signature after GSR (*Figure 2—figure supplement 4B*, right). In contrast, scans whose SLFOs-neural similarity was close to zero after GSR (*Figure 2—figure supplement 4C*) did not exhibit increased FC in the somatosensory and auditory networks (*Figure 2—figure supplement 4D*). Thus, the different levels of correlation in these two networks, as compared to other networks, were likely responsible for the negative GS-SLFOs similarity after GSR. A possible explanation for the low contribution of SLFOs in the somatosensory and auditory networks is the use of a single global regressor for the estimation of the SLFOs signature, which does not account for variability in the dynamics of SLFOs across brain regions. Interestingly, the primary sensory areas that belong to the two aforementioned networks were also reported in the work by *Chen et al., 2020* as exhibiting a different respiration response function (RRF) compared to other areas. These findings suggest that modeling SLFOs at the individual level could potentially be improved if spatial variability in RRF is also accounted for. Moreover, these results indicate that the effects of SLFOs cannot be corrected equally well in all areas solely by regressing out the GS. It should be noted, though, that the RRF spatial variability reported by *Chen et al., 2020* was observed at the group level using a relatively flexible model with 10 free parameters. Future studies are still needed to address how the SLFOs can be modeled at the individual level in a manner that takes into consideration both subject and spatial specificity, while avoiding overfitting. Related to this, we also observed that the negative SLFOs-neural similarity was substantially decreased when GSR is combined with white matter denoising, which may suggest that the nuisance regressors obtained from white matter can reduce fluctuations induced by SLFOs in all areas, including primary sensory areas, better than GSR alone.

Regarding time-varying FC, GSR did not reduce the effect of nuisance processes equally well compared to static FC (*Figure 2E–F* vs. *Figure 5A–B*). Nonetheless, a recent study evaluating pre-processing strategies in the context of time-varying FC showed that incorporating GSR in the pre-processing improved the identification of modularity in functional networks (*Lydon-Staley et al.,*

*2019*). This may indicate that GSR was able to remove nuisance processes that we did not evaluate in the current study. These processes may be related to scanner instabilities, $CO_2$ concentration (*Power et al., 2017*; *Wise et al., 2004*) and finger skin vascular tone (*Kassinopoulos and Mitsis, 2021b*; *Özbay et al., 2019*), which are known to be reflected on the GS.

Despite the effectiveness of GSR in reducing nuisance confounds from the data, we cannot exclude the possibility of removing some neuronal-related fluctuations. Alternatives to GSR that have been proposed to remove global artifacts include time delay analysis using 'rapidtide' (*Tong et al., 2019*), removal of the first principal component from the fMRI data (*Carbonell et al., 2011*), removal of fluctuations associated to large clusters of coherent voxels (*Aquino et al., 2020*), and the use of temporal ICA (*Glasser et al., 2018*), albeit the latter is only applicable to datasets with a large number of subjects such as the HCP.

## The effect of phase encoding direction in connectivity

Earlier studies have demonstrated that chest wall movements due to breathing perturb the $B_0$ field (*Raj et al., 2001*; *Raj et al., 2000*; *Van de Moortele et al., 2002*), which has consequences on EPI fMRI data. While this phenomenon is not fully understood, it seems to have two main effects that are observable along the phase encoding direction: (1) Breathing causes factitious motion of the fMRI volumes in the phase encoding direction (*Raj et al., 2001*; *Raj et al., 2000*). This effect has sparkled attention recently, since it has been recognised that it may have critical implications for motion correction when performing censoring (i.e. removal of motion-contaminated fMRI volumes) in multi-band (*Fair et al., 2020*; *Power et al., 2019*) and single-band (*Gratton et al., 2020*) data. (2) Breathing induces artifacts on voxel timeseries that depend on the location of those voxels along the phase encoding direction (*Raj et al., 2001*; *Raj et al., 2000*). Our results provide further evidence in support of the latter effect. Specifically, we found that depending on the phase encoding direction (LR or RL), breathing motion artifacts were more pronounced in the left or right hemisphere respectively (*Figure 1C*). Moreover, we observed that breathing motion increased within-hemisphere connectivity for both phase encoding scan types (*Figure 2C*, *Figure 2—figure supplement 1C*), which implies that breathing induces artifactual fluctuations that are to a certain extent different between hemispheres. However, note that the connectome profile of breathing motion exhibited some differences between the two phase encoding directions (*Figure 2—figure supplement 1C*), which explains the higher connectome fingerprinting accuracy in the breathing motion dataset when examining pairs of scans with the same phase encoding direction, compared to scans with different phase encoding direction (*Figure 3A*).

Our results point to a systematic effect of breathing on static FC through variations in the $B_0$ magnetic field. Importantly, this systematic bias is contingent on the phase encoding direction, which seems to indicate that factitious rather than real motion is the predominant source of respiration-related motion artifacts in fMRI, as has been previously suggested (*Brosch et al., 2002*; *Raj et al., 2001*). Even though common preprocessing pipelines greatly reduce these effects, they do not eliminate them (*Figure 2G*). Thus, studies that consider datasets with different phase encodings, should be aware of the effect of phase encoding on FC, especially if data from different groups have been acquired with different phase encodings.

## Individual discriminability

Test-retest reliability is important for establishing the stability of inter-individual variation in fMRI FC across time. However, apart from neural processes, nuisance processes can also have an impact on test-retest reliability, given the subject-specific nature of physiological processes (*Batchvarov et al., 2002*; *Golestani et al., 2015*; *Malik et al., 2008*; *Pinna et al., 2007*; *Pitzalis et al., 1996*; *Power et al., 2020*; *Reland et al., 2005*) and head motion (*Van Dijk et al., 2012*; *Zeng et al., 2014*). This leads to the concerning notion that nuisance processes may be artifactually driving the reports of high reliability in FC measures. For instance, it has been reported that the median of intra-class correlation values across functional connections, which is a metric of test-retest reliability, is reduced when a relatively aggressive pipeline is used (*Birn et al., 2014*; *Parkes et al., 2018*). Furthermore, motion can classify subjects at above-chance levels (*Horien et al., 2019*), and breathing motion is more prominent in older individuals and those with a higher body mass index

(*Gratton et al., 2020*). In the present study, we examined the potential effect of nuisance processes on subject discriminability using connectome fingerprinting.

## Whole-brain identification

To assess the individual discriminability of nuisance processes, we performed connectome fingerprinting analysis using the generated nuisance datasets. All nuisance processes exhibited identification accuracy above chance level (*Figure 3A*). Pairs of scans with the same phase encoding yielded higher identification accuracy than pairs of scans with different phase encoding (*Figure 3A*). This effect is particularly evident for breathing motion, and to a lesser extent cardiac pulsatility and head motion. This observation suggests that not only these confounds exert a distinctive artifactual spatial pattern that is dependent on the phase encoding direction, which can be also observed upon careful examination of *Figure 1B–D*, but also that this artifactual pattern is to a certain degree subject-specific. On the other hand, the subject discriminability of SLFOs is not modulated by phase encoding (*Figure 3A*). Given the nature of SLFOs (i.e. they affect the BOLD signal through changes in CBF), the high subject discriminability of SLFOs suggests a certain degree of idiosyncrasy that is possibly related to the vascular architecture of an individual. Overall, our results suggest that there is some degree of subject discriminability in nuisance processes.

Identification accuracies of 'neural' datasets were very high for all preprocessing strategies (*Figure 3B*), in line with previous studies (*Finn et al., 2015*; *Horien et al., 2019*). WM denoising, which was found to be the most effective strategy for reducing confounds due to head motion and physiological fluctuations (*Figure 2E–H*), yielded also the highest accuracy in connectome fingerprinting (*Figure 3B*), suggesting that the high subject discriminability observed in the HCP data is not due to the presence of confounds. Interestingly, the increased accuracy observed in the nuisance datasets for scans with the same phase encoding (*Figure 3A*) was also observed in the case of the raw data (*Figure 3B*). In contrast, for the rest of the pipelines the difference in accuracy between pairs of scans from different days with the same or different phase encoding direction vanishes (*Figure 3B*). This is likely because of the reduction of nuisance effects, mainly breathing motion artifacts. Note also that for both mild and aggressive pipelines, pairs of scans from the same day exhibited higher accuracies compared to pairs of scans from different days, which cannot be attributed to potential residuals of nuisance fluctuations (*Figure 3A*). Possible explanations for this finding are that the functional connectome of a subject reflects some aspects of their vigilance levels (*Tagliazucchi and Laufs, 2014*; *Thompson et al., 2013a*; *Wang et al., 2016*), mind-wandering (*Gonzalez-Castillo et al., 2019*; *Gorgolewski et al., 2014*; *Kucyi, 2018*; *Kucyi and Davis, 2014*), or the effect of time of day (*Hodkinson et al., 2014*; *Jiang et al., 2016*; *Orban et al., 2020*; *Shannon et al., 2013*), which can differ across sessions. Overall, the high connectome-based identification accuracies reported in the literature do not appear to be driven by nuisance confounds, suggesting a neural origin underpinning the inter-individual differences in connectivity. Nonetheless, it is worth pointing out that subject variability in the magnitude of functional connections has been shown to arise as a result of spatial topographical variability in the location of functional regions across individuals (*Bijsterbosch et al., 2018*), which could also explain the high subject discriminability observed in fMRI-based connectomes.

## Network-based identification

We observed that edges within association cortices (e.g. parts of the frontoparietal, default mode, and cinguloopercular systems) exhibited the highest subject specificity (*Figure 4B*), consistent with previous studies (*Finn et al., 2015*; *Gratton et al., 2018*; *Horien et al., 2019*; *Mueller et al., 2013*; *Seitzman et al., 2019*; *Vanderwal et al., 2017*). The fact that association cortices are the most evolutionarily recent (*Zilles et al., 1988*) and are thought to be involved in higher level functions (*Cole et al., 2014*; *Cole et al., 2013*; *Dosenbach et al., 2007*; *Gratton et al., 2017*; *Raichle, 2015*) has been posited as a possible reason for the high identification accuracy yielded by these networks. On the other hand, it has also been speculated that medial frontal and frontoparietal networks exhibit the highest identification accuracy as a result of being less prone to distortions from susceptibility artifacts (*Horien et al., 2019*; *Noble et al., 2017*). If the latter was the case, we would expect to see decreased accuracy for these networks when probing the nuisance datasets. However, we did not observe such a tendency for any of the nuisance processes evaluated (*Figure 4A*), and

preprocessing strategies that successfully removed nuisance processes yielded enhanced subject discriminability of control networks and the DMN (*Figure 4C*). These results seem to indicate that the basis of the high identification rates for association cortices is of neural origin, and thus that resting-state fMRI-based connectome fingerprinting can capture idiosyncratic aspects of cognition reflected on the resting-state functional characteristics of the association cortex.

## Conclusions

The current study introduces a novel framework for assessing the effects of the main fMRI confounds on static and time-varying FC. Our results suggest that head motion and systemic BOLD fluctuations associated to changes in heart rate and breathing patterns cause systematic biases in static FC and result in recurrent patterns in time-varying FC. Data-driven techniques based on decomposing the data into principal or independent components (PCA, ICA), combined with GSR, lead to the strongest reduction of the aforementioned effects. Importantly, these preprocessing strategies also improve connectome-based subject identification, indicating that the high subject discriminability reported in the literature is not attributable to nuisance processes.

# Materials and methods

## Human Connectome Project (HCP) dataset

The resting-state fMRI data analysed in this study are from the S1200 release of the 3T HCP dataset (*Smith et al., 2013*; *Van Essen et al., 2013*), which consists of young, healthy twins and siblings (age range: 22–36 years). The HCP dataset includes, among others, resting-state data acquired on 2 different days, during which subjects were instructed to keep their eyes open and fixated on a cross-hair. Each day included two consecutive 15 min resting-state runs, acquired with left-to-right (LR) and right-to-left (RL) phase encoding direction. During each fMRI run, 1200 frames were acquired using a gradient-echo echo-planar imaging (EPI) sequence with a multiband factor of 8, spatial resolution of 2 mm isotropic voxels, and a TR of 0.72 s. Further details of the data acquisition parameters can be found in previous publications (*Smith et al., 2013*; *Van Essen et al., 2012*). Concurrently with fMRI images, cardiac and respiratory signals were measured using a standard Siemens pulse oximeter placed on the fingertip and a breathing belt placed around the chest, with a 400 Hz sampling rate.

We only considered subjects who had available data from all four runs, and excluded subjects based on the quality of the physiological recordings (see section 5.2.1 below for details). Pulse oximeter and respiratory belt signals from ~1000 subjects were first visually inspected to determine their quality, since their traces are often not of sufficient quality for reliable peak detection (*Power, 2019*). The selection process resulted in a final dataset with 392 subjects (ID numbers provided in Supp. Material).

## Preprocessing

### Preprocessing of physiological recordings

After selecting subjects with good quality traces, the pulse wave was processed to automatically detect beat-to-beat intervals (RR), and the heart rate signal was further computed as the inverse of the time differences between pairs of adjacent peaks and converted to units of beats-per-minute (bpm). Heart rate traces were visually checked to ensure that outliers and abnormalities were not present. An outlier replacement filter was used to eliminate spurious changes in heart rate when these changes were found to be due to sporadic noisy cardiac measurements (for more details see Supp. Figures 1 and 2 from *Kassinopoulos and Mitsis, 2019*). We also excluded subjects with a heart rate of exactly 48 bpm and lack of heartbeat interval variability, as they have been pointed out as outliers in recent studies (*Orban et al., 2020*; *Valenza et al., 2019*). The signal from the breathing belt was detrended linearly, visually inspected and corrected for outliers using a replacement filter. Subsequently, it was low-pass filtered at 5 Hz and *Z*-scored. The respiratory flow, proposed in *Kassinopoulos and Mitsis, 2019* as a robust measure of the absolute flow of inhalation and exhalation of a subject at each time point, was subsequently extracted by applying further smoothing on the breathing signal (moving average filter of 1.5 s window) and, subsequently, computing the

square of the derivative of the smoothed breathing signal. Finally, heart rate and respiratory flow time-series were re-sampled at 10 Hz.

An example code (Preprocess_Phys.m) showing the detailed specifications of the algorithms used during the preprocessing of the physiological signals is available on github.com/mkassinopoulos/PRF_estimation/.

## Preprocessing of fMRI data: assessing the impact of nuisance correction strategies

From the HCP database we downloaded the minimally preprocessed data described in *Glasser et al., 2013* and the FIX-denoised data, both in volume and surface space. Briefly, the minimal preprocessing pipeline included removal of spatial distortion, motion correction via volume re-alignment, registration to the structural image, bias-field correction, 4D image intensity normalization by a global mean, brain masking, and non-linear registration to MNI space. Further steps to obtain surface data were volume to surface projection, multimodal inter-subject alignment of the cortical surface data (*Robinson et al., 2014*), and 2 mm (FWHM) surface-constrained smoothing. Additional steps following minimal preprocessing to obtain the FIX-denoised data were de-trending using a mild high-pass filter (2000 s), head motion correction via 24 parameter regression, and denoising via spatial ICA followed by an automated component classifier (FMRIB's ICA-based X-noiseifier, FIX) (*Griffanti et al., 2014*; *Salimi-Khorshidi et al., 2014*). Minimal spatial smoothing (FWHM = 4 mm) was applied to the downloaded minimally preprocessed and FIX-denoised volumetric data. Both minimally preprocessed and FIX-denoised data were parcellated employing the Gordon atlas across 333 regions of interest (ROIs) (*Gordon et al., 2016*) and the Seitzman atlas across 300 ROIs (*Seitzman et al., 2020*), using the surface and volume space data, respectively. ROIs that did not belong to a brain network were disregarded, hence a total of 286 ROIs (Gordon atlas) and 285 ROIs (Seitzman atlas) were retained for further analyses. The main differences between these two brain parcellations, apart from being computed on the surface and volume space respectively, are that the ROIs in the Gordon atlas do not have the same size, whereas in the Seitzman atlas the ROIs are all spheres of 8 mm diameter, and that the Gordon atlas only includes cortical regions, whereas the Seitzman atlas includes cortical and subcortical regions. The results from the Gordon atlas are presented in the main manuscript whereas the results from the Seitzman atlas can be found in the Supplementary Material (*Figure 2—figure supplement 5*, *Figure 3—figure supplement 2*, *Figure 5—figure supplement 2*). Further, the parcellated data were high-pass filtered at 0.01 Hz.

In addition to the FIX-denoising strategy, several other data-driven preprocessing techniques were evaluated to assess the extent to which they were able to remove physiological and motion-driven confounds (*Table 1*). We chose pipelines that had been used in the landmark FC studies of *Finn et al., 2015* and *Laumann et al., 2017*. These were denoted as 'mild' pipelines, since they regress out considerably fewer components compared to FIX. Further, we also included two more aggressive pipelines that were found to outperform previously proposed techniques in terms of network identifiability (*Kassinopoulos and Mitsis, 2021a*). Nuisance regression was performed after the minimally preprocessed data had been parcellated to reduce computational time. All

**Table 1.** Preprocessing strategies examined.
All strategies were evaluated with and without global signal regression (GSR).

| Preprocessing strategy | Acronym | Regressors included |
|---|---|---|
| Minimally preprocessed HCP data | Raw data | None |
| FIX-denoised HCP data | FIX | ICA-based technique combined with an automated component classifier (*Salimi-Khorshidi et al., 2014*) |
| Pipeline used in *Finn et al., 2015*, *Nature Neuroscience* | MildA | Mean time-series of the white matter and CSF voxels (2); realignment parameters and their first derivatives (12) |
| Pipeline used in *Laumann et al., 2017*, *Cerebral Cortex* | MildB | Mean time-series of the white matter and CSF voxels and their derivatives (4); realignment parameters and their first derivatives, quadratic terms, and squares of derivatives (24) |
| Pipeline proposed by *Kassinopoulos and Mitsis, 2021a*, *bioRxiv* | $WM_{50}$ | 50 PCA components from white matter voxels |
| | $WM_{200}$ | 200 PCA components from white matter voxels |

preprocessing strategies were evaluated with and without global signal regression (GSR), since the latter is still somewhat controversial (*Liu et al., 2017*; *Murphy et al., 2009*; *Murphy and Fox, 2017*). To facilitate the comparison between preprocessing strategies, the minimally preprocessed data were also evaluated, yielding in total 12 preprocessing strategies. Given that the minimal pre-processing pipeline consists of only the initial steps for fMRI denoising, for simplicity in the results we refer to these data as raw data. The regressors included in each preprocessing strategy can be found in *Table 1*. Note that for the pipeline from *Laumann et al., 2017* the derivative of the global signal was also regressed out. The global signal for the surface and volumetric data was computed as the average fMRI timeseries across vertices and the whole brain respectively.

## Nuisance processes evaluated

The following four nuisance processes were considered (*Table 2*):

### Systemic low frequency oscillations (SLFOs)

SLFOs refer to non-neuronal global BOLD fluctuations. Major sources of SLFOs are spontaneous fluctuations in the rate or depth of breathing (*Birn et al., 2006*) and fluctuations in heart rate (*Chang et al., 2009*; *Shmueli et al., 2007*). The former mainly exert their effects via changing the concentration of arterial $CO_2$, which is a potent vasodilator, altering CBF and thus the BOLD fMRI signal (*Birn et al., 2008a*; *Birn et al., 2008b*; *Chang and Glover, 2009b*; *Prokopiou et al., 2019*; *Wise et al., 2004*), Importantly, there is evidence that SLFOs are a more substantial source of physi-ological noise in BOLD fMRI compared to high frequency cardiac pulsatility and breathing motion artifacts (*Tong et al., 2019*; *Tong and Frederick, 2014*). In this study, SLFOs were modeled follow-ing a framework proposed in our previous work (*Kassinopoulos and Mitsis, 2019* scripts available on github.com/mkassinopoulos/PRF_estimation/). Briefly, the extracted heart rate and respiratory flow signals were fed into an algorithm that estimated scan-specific physiological response functions (PRFs). This algorithm estimates PRF curves that maximize the correlation between their convolution with heart rate and respiratory flow and the global signal of the same scan, while ensuring that the shapes of the PRF curves are physiologically plausible. The heart rate and respiratory flow signals were subsequently convolved with their respective PRFs and added to each other, yielding time-series that reflect the total effect of SLFOs (*Figure 7—figure supplement 1A*). These time-series were used in the current study as the physiological regressor related to SLFOs.

### Cardiac pulsatility

Pulsatility of blood flow in the brain can cause pronounced modulations of the BOLD signal (*Noll and Schneider, 1994*), which tend to be localized along the vertebrobasilar arterial system and the sigmoid transverse and superior sagittal sinuses (*Dagli et al., 1999*; *Kassinopoulos and Mitsis, 2019*). We modeled fluctuations induced by cardiac pulsatility using six regressors obtained with 3rd order RETROICOR (*Glover et al., 2000*), based on the pulse oximeter signal of each scan.

### Breathing Motion

Chest movements during the breathing cycle generate head motion in the form of head nodding by mechanical linkage through the neck, but also factitious motion (also known as pseudomotion)

**Table 2.** Nuisance processes examined.

|  | SLFOs | Cardiac pulsatility | Breathing motion | Head motion |
|---|---|---|---|---|
| Recordings | Pulse oximeter, breathing belt, fMRI global signal | Pulse oximeter | Breathing belt | fMRI |
| Signals employed | Heart rate, respiratory flow, global signal | Cardiac cycle | Breathing cycle | Realignment parameters and derivatives |
| Model | *Kassinopoulos and Mitsis, 2019*, *NeuroImage* | 3rd order RETROICOR *Glover et al., 2000*, *NeuroImage* | 3rd order RETROICOR *Glover et al., 2000*, *NeuroImage* | - |
| Number of regressors | 1 | 6 | 6 | 12 |

through small perturbations of the $B_0$ magnetic field caused by changes in abdominal volume when air enters the lungs (*Power et al., 2019*; *Raj et al., 2001*; *Van de Moortele et al., 2002*). We modeled breathing-induced fluctuations using six regressors obtained with 3rd order RETROICOR (*Glover et al., 2000*), based on the respiratory signal of each scan.

## Head motion

Subject motion produces substantial signal disruptions in fMRI studies (*Friston et al., 1996*; *Power et al., 2012*) and is a major confound when evaluating connectivity differences between groups with dissimilar tendencies for motion (*Makowski et al., 2019*; *Satterthwaite et al., 2012*; *Van Dijk et al., 2012*). We quantified head motion using the six realignment parameters as well as their temporal first derivatives provided by the HCP. The six physiological regressors related to breathing motion were regressed out from the realignment parameters and their derivatives, since true and factitious motion due to breathing is reflected on the realignment parameters (*Fair et al., 2020*).

All nuisance regressors were high-pass filtered at 0.01 Hz to ensure similar spectral content with the fMRI data and thus avoid reintroduction of nuisance-related variation (*Bright et al., 2017*; *Hallquist et al., 2013*). The regressors were then normalized to zero mean and unit variance. *Figure 7—figure supplement 2* demonstrates the spectral content of each nuisance process, as well as the effect of regressing out breathing motion from the realignment parameters. Physiological and motion traces from three illustrative scans, along with fMRI carpet plots, can be found in *Figure 7—figure supplement 3*. Nuisance traces and carpet plots from all subjects are available on figshare (*Xifra-Porxas et al., 2021b*).

## Isolation of nuisance fluctuations from fMRI data

We propose a framework to isolate nuisance fluctuations for each of the aforementioned processes, which reflects the physiologically-driven fluctuations and head motion artifacts observed in the fMRI data (*Figure 7*). A similar methodology was used in *Bright and Murphy, 2015* to investigate whether preprocessing strategies remove variance associated to resting-state networks.

Initially, the contribution of each nuisance process on the ROI time-series was quantified using a generalised linear model, formulated as:

$$y(t) = \beta_0 + \beta_{SLFOs}x_{SLFOs}(t) + \beta_{CP}\mathbf{x}_{CP}(t) + \beta_{BM}\mathbf{x}_{BM}(t) + \beta_{HM}\mathbf{x}_{HM}(t) + \varepsilon(t) \tag{1}$$

$$\beta_{CP} = \begin{bmatrix} \beta_{CP}^1 & \cdots & \beta_{CP}^6 \end{bmatrix}, \beta_{BM} = \begin{bmatrix} \beta_{BM}^1 & \cdots & \beta_{BM}^6 \end{bmatrix}, \beta_{HM} = \begin{bmatrix} \beta_{HM}^1 & \cdots & \beta_{HM}^{12} \end{bmatrix}$$

$$\mathbf{x}_{CP}(t) = \begin{bmatrix} x_{CP}^1(t) \\ \vdots \\ x_{CP}^6(t) \end{bmatrix}, \mathbf{x}_{BM}(t) = \begin{bmatrix} x_{BM}^1(t) \\ \vdots \\ x_{BM}^6(t) \end{bmatrix}, \mathbf{x}_{HM}(t) = \begin{bmatrix} x_{HM}^1(t) \\ \vdots \\ x_{HM}^{12}(t) \end{bmatrix}$$

where $y$ are ROI time-series from the minimally preprocessed data, $x_{SLFOs}$ is the physiological regressor modeling SLFOs, $\mathbf{x}_{CP}(t)$ are the 6 physiological regressors modeling cardiac pulsatility, $\mathbf{x}_{BM}(t)$ are the 6 physiological regressors modeling breathing motion, $\mathbf{x}_{HM}(t)$ are the 12 regressors modeling head motion, $\{\beta_0, \beta_{SLFOs}, \beta_{CP}, \beta_{BM}, \beta_{HM}\}$ denote the parameters to be estimated, and $\varepsilon$ is the error (or residual). As can be seen, all four nuisance processes (25 regressors in total) were included simultaneously in the regression to model the BOLD signal fluctuations in a specific ROI. Note that no significant collinearity was observed between the 25 regressors (*Figure 7—figure supplement 4*).

For each nuisance process, the estimated values $\hat{\beta}$ were multiplied by their corresponding regressors and added together to obtain the fluctuations of the nuisance process of interest ($\hat{y}_{NPI}(t)$) within a specific ROI, and a 'clean' time-series was calculated via removal of all other nuisance processes ($\hat{y}_{NPI+Neur}(t)$), as follows:

$$\text{SLFOs}: \quad \hat{y}_{NPI}(t) = \hat{\beta}_{SLFOs}x_{SLFOs}(t) \tag{2}$$

$$\hat{y}_{NPI+Neur}(t) = y(t) - \hat{\beta}_0 - \hat{\beta}_{CP}\mathbf{x}_{CP}(t) - \hat{\beta}_{BM}\mathbf{x}_{BM}(t)\hat{\beta}_{HM}\mathbf{x}_{HM}(t) \tag{3}$$

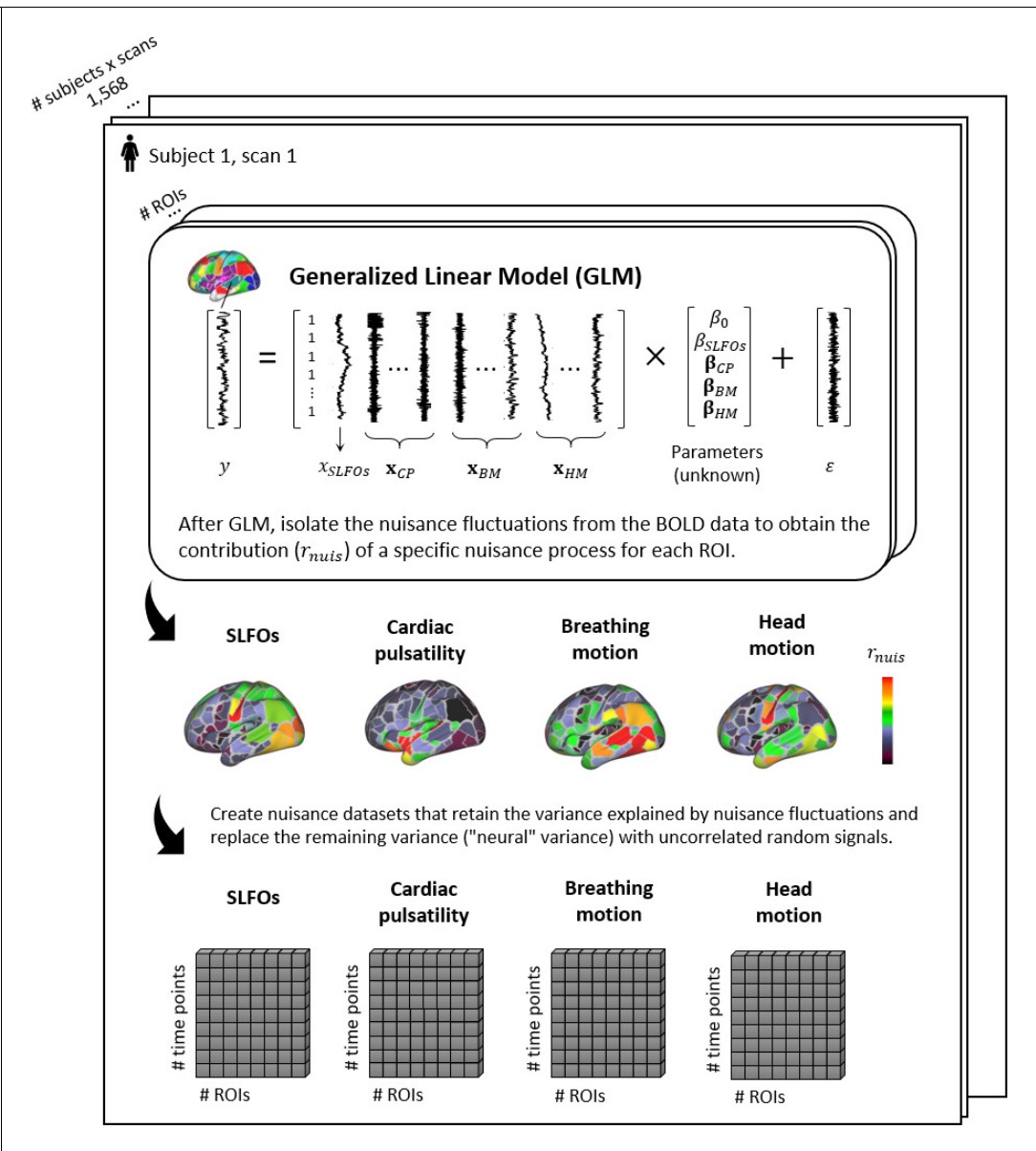

**Figure 7.** Graphical summary of the proposed framework for isolating the fluctuations for each physiological process. For each scan, the ROI time-series are modeled using the regressors related to systemic low frequency oscillations (SLFOs), cardiac pulsatility (CP), breathing motion (BM), and head motion (HM). Subsequently, the fraction of BOLD variance explained by each nuisance process is isolated and employed to generate synthetic datasets that only contain nuisance fluctuations.

The online version of this article includes the following figure supplement(s) for figure 7:

**Figure supplement 1.** Illustration of methodology for modeling systemic low frequency oscillations (SLFOs) using the traces of heart rate and breathing activity.

**Figure supplement 2.** Power spectral densities of the nuisance processes evaluated in the present study.

**Figure supplement 3.** Nuisance traces from three illustrative scans.

**Figure supplement 4.** Collinearity across nuisance regressors.

**Figure supplement 5.** Histogram of autoregressive coefficients.

$$\text{Cardiac pulsatility}: \quad \hat{y}_{NPI}(t) = \hat{\beta}_{CP}\mathbf{x}_{CP}(t) \tag{4}$$

$$\hat{y}_{NPI+Neur}(t) = y(t) - \hat{\beta}_0 - \hat{\beta}_{SLFOs}x_{SLFOs}(t) - \hat{\beta}_{BM}\mathbf{x}_{BM}(t) - \hat{\beta}_{HM}\mathbf{x}_{HM}(t) \tag{5}$$

$$\text{Breathing motion}: \ \hat{y}_{NPI}(t) = \hat{\beta}_{BM}\mathbf{x}_{BM}(t) \tag{6}$$

$$\hat{y}_{NPI+Neur}(t) = y(t) - \hat{\beta}_0 - \hat{\beta}_{SLFOs}x_{SLFOs}(t) - \hat{\beta}_{CP}\mathbf{x}_{CP}(t) - \hat{\beta}_{HM}\mathbf{x}_{HM}(t) \tag{7}$$

$$\text{Head motion}: \ \hat{y}_{NPI}(t) = \hat{\beta}_{HM}\mathbf{x}_{HM}(t) \tag{8}$$

$$\hat{y}_{NPI+Neur}(t) = y(t) - \hat{\beta}_0 - \hat{\beta}_{SLFOs}x_{SLFOs}(t) - \hat{\beta}_{CP}\mathbf{x}_{CP}(t) - \hat{\beta}_{BM}\mathbf{x}_{BM}(t) \tag{9}$$

In this manner, we generated 'cleaned' ROI time-series ($\hat{y}_{NPI+Neur}(t)$) in which all considered noisy fluctuations were removed except the ones corresponding to the specific nuisance process being evaluated. The next step was to quantify the contribution of the latter to the remaining fluctuations within each ROI. To achieve this, the estimated nuisance signal was correlated to the 'clean' ROI time-series:

$$r_{nuis} = \text{corr}(\hat{y}_{NPI}(t), \hat{y}_{NPI+Neur}(t)) \tag{10}$$

Subsequently, the estimated nuisance signal was subtracted from the 'clean' ROI time-series to obtain what is typically considered the 'neural' time-series. These time-series were correlated to the 'clean' time-series to quantify the contribution of the 'neural' variations to the total ROI signal fluctuations:

$$\hat{y}_{Neur}(t) = \hat{y}_{NPI+Neur}(t) - \hat{y}_{NPI}(t) \tag{11}$$

$$r_{neur} = \text{corr}(\hat{y}_{Neur}(t), \hat{y}_{NPI+Neur}(t)) \tag{12}$$

Afterwards, nuisance datasets for each process were created by scaling the estimated nuisance signal within each ROI with its corresponding correlation coefficient $r_{nuis}$ and a first order autoregressive (AR(1)) process ($\psi(t)$) scaled with r_neur. We used an AR(1) model as it has been shown to be able to capture both the static and time-varying FC structure of resting-state fMRI data (*Liégeois et al., 2017*). This is expressed as:

$$y_{Nuis}(t) = r_{nuis}Z[\hat{y}_{NPI}(t)] + r_{neur}Z[\psi(t)] \tag{13}$$

$$\psi(t) = a_1\psi(t-1) + \xi(t) \tag{14}$$

where $Z[\cdot]$ denotes normalization to zero mean and unit variance. The coefficient $a_1$ was randomly sampled from a distribution of coefficients generated through fitting an AR(1) model to the real fMRI data (*Figure 7—figure supplement 5*).

Thus, this framework generated four synthetic nuisance datasets that contained the isolated fluctuations from each of the nuisance processes evaluated. In a sense, the ROI time-series in each nuisance dataset are equivalent to the term $\hat{y}_{NPI+Neur}(t)$, with the 'neural' fluctuations replaced by random autocorrelated processes. These time-series were used to characterize the connectome profile of the nuisance processes without the presence of neurally related signals, while maintaining the noise-to-signal ratio between physiological/motion-related noise and 'neural' signal intact. Note that if the nuisance datasets consisted solely of artifactual fluctuations without the AR(1) process added, this would result in an overestimation of the correlation fraction attributed to the nuisance processes that was present in the experimental fMRI data. This can be easily understood in the case of two ROI time-series that are weakly driven by SLFOs. As the contribution of this nuisance process to the aggregate ROI time-series would be small, the parameter $\beta_{SLFOs}$ in *Equation 2* for both ROIs would be relatively small as well. However, without the addition of autocorrelated processes, the correlation of the time-series $y_{NPI}$ associated with these two ROIs would be 1.00 (or $-1.00$ depending on the signs of the corresponding beta parameters) as Pearson's correlation is a metric that is blind to the variance of the signals, thereby overestimating the contribution of SLFOs to the FC between those ROIs.

## Estimation of static and time-varying functional connectivity (FC)

Using the pre-processed fMRI data from each of the 12 pipelines (see section 5.2.2), henceforth called 'neural' datasets, and the 4 nuisance datasets, we performed Pearson correlation analyses between brain regions over the whole scan (static FC) and within sliding windows (time-varying FC). To quantify time-varying FC, the entire scan was split up into 62 sliding windows of 43.2 sec (60 samples) duration, with 70% overlap in time. Subsequently, for each scan, we computed the functional connectivity dynamics (FCD) matrix (*Hansen et al., 2015*), which is a symmetric matrix in which the entries $(i, j)$ correspond to the Pearson correlations between the upper triangular elements of the FC matrices in windows $i$ and $j$. The size of the FCD matrix was $W \times W$, where $W$ is the number of windows (62). Thus, while the static FC matrix characterizes the spatial structure of resting activity, the FCD matrix captures the temporal evolution of connectome correlations.

The analyses resulted in 32 matrices for each subject (*Figure 8*): four static FC and 4 FCD nuisance matrices (one for each physiological process considered), as well as 12 static FC and 12 FCD 'neural' matrices (one for each preprocessing strategy evaluated). To quantify the influence of the nuisance processes on static FC and FCD for each preprocessing strategy, similarities between pairs of nuisance and 'neural' matrices were evaluated by correlating their upper triangular values. Note that for the FCD matrices, upper triangular elements corresponding to the correlation between

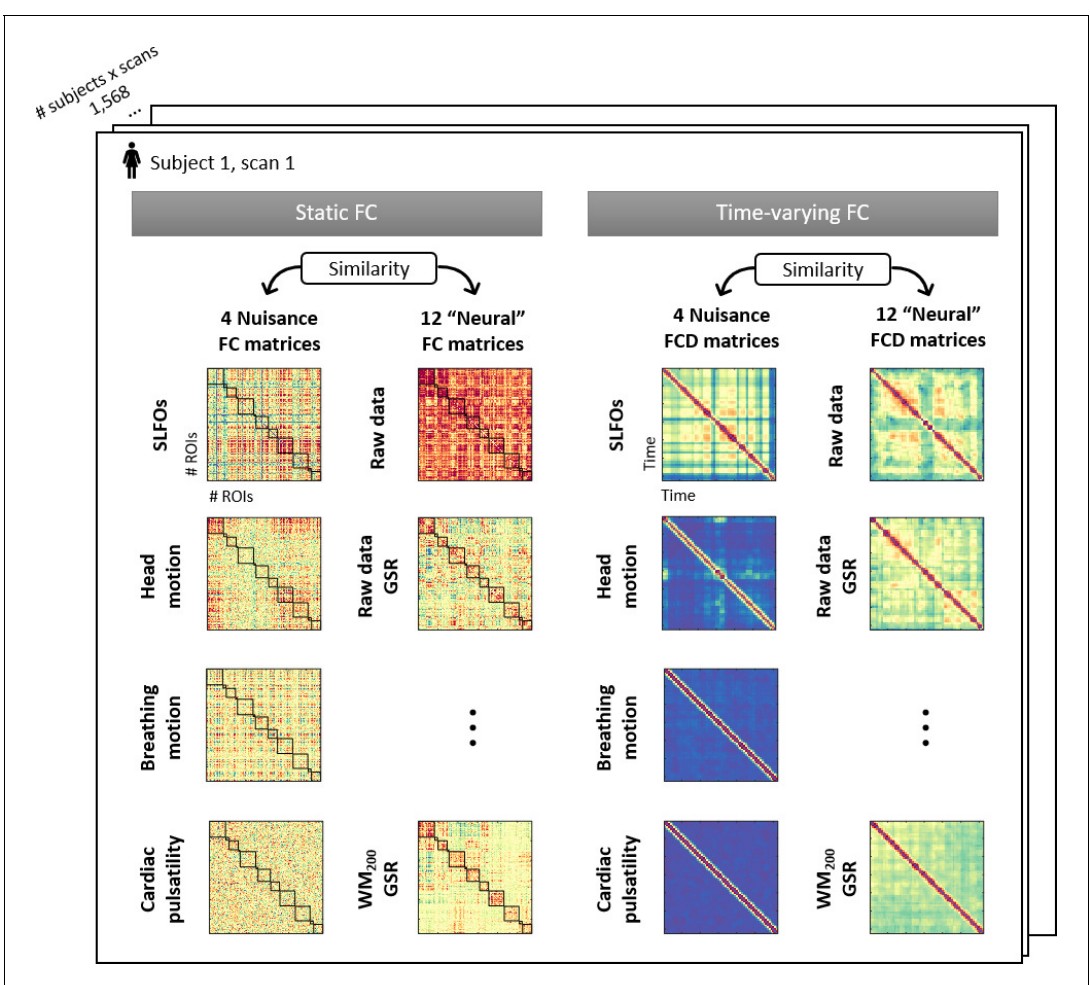

**Figure 8.** Illustration of the 32 connectivity matrices computed per scan. For both static and time-varying FC analyses, four nuisance connectivity matrices were computed using the generated nuisance datasets, as well as 12 'neural' connectivity matrices corresponding to the 12 pre-processing strategies evaluated (note that each of the six strategies listed in *Table 1* was assessed with and without GSR). Static FC matrices were computed as the correlation across brain regions using the whole scan. Time-varying FC analysis constructed time-resolved connectivity matrices as the correlation between static FC matrices within sliding windows, known as functional connectivity dynamics (FCD) matrices (*Hansen et al., 2015*).

overlapping windows were disregarded because of high correlation by design (see block diagonal in *Figure 8*). All correlation values were Fisher z transformed.

## Connectome-based identification of individuals

We implemented a connectome-based identification of individual subjects using the static FC matrices to investigate the potential effect of 'nuisance fingerprints' on the degree of subject specificity in individual FC metrics. The identification procedure, known as connectome fingerprinting, has been described in detail previously (*Finn et al., 2015*). Briefly, a database was first created consisting of all subjects' FC matrices from a particular resting-state scan (*Figure 3—figure supplement 1B*). An FC matrix from a specific subject and different resting-state scan was then selected and denoted as the target ($Subj_x$). Pearson correlation coefficients ($r_1, \ldots, r_N$) were computed between the upper triangular values of the target FC matrix and all the FC matrices in the database. If the highest correlation coefficient corresponded to a pair of FC matrices from the same subject, a successful identification was indicated ($ID_{Subj_x} = 1$); otherwise, it was marked as an incorrect identification ($ID_{Subj_x} = 0$). The identification test was repeated such that each subject serves as the target subject once, and then the ID values were averaged across subjects to obtain the identification accuracy of the database-target pair. This process was repeated until tests between all scanning sessions were performed. In total, 12 database-target combinations were computed (*Figure 3—figure supplement 1A*). Identification was performed using the whole brain connectivity matrix, as well as based on edges from within and between networks. In the latter case, networks containing less than 10 ROIs were excluded from the analysis.

The connectome fingerprinting analysis was performed independently for each physiological process, as well as for each preprocessing strategy, using the generated FC matrices (*Figure 8*). This analysis was only performed on static FC matrices and not time-varying FC matrices because recurrent patterns of connectivity observed in the FCD matrices are not expected to occur at similar time instances between scans.

## Statistics

To assess the significance of the results, surrogate nuisance datasets were generated via inter-subject surrogates (*Lancaster et al., 2018*), using fMRI data recorded from one subject's scan and physiological signals recorded from a different subject's scan (in the case of the head motion dataset, volume realignment parameters were employed). Note that when creating each surrogate dataset, only the nuisance process being examined was replaced by signals from a different subject, whereas all other nuisance regressors remained the same. This procedure was repeated 1000 times for each nuisance process, where each surrogate dataset consisted of permuting the nuisance signals across subjects. A null distribution for each brain region was computed by estimating the mean contribution to BOLD across subjects for each surrogate nuisance dataset, leading to a distribution of 1000 values. Subsequently, the significance of the nuisance contributions to the BOLD signal were assessed by comparing the observed contribution (mean across subjects) to the corresponding null distribution for each ROI. The significance level at $p < 0.05$ was corrected for multiple comparisons using false discovery rate (FDR). Furthermore, the similarity between the nuisance and 'neural' FC matrices was compared against the similarity obtained using surrogate nuisance FC matrices. Note that for visualization purposes, similarity values identified as outliers (> 3 SD) are not displayed in *Figure 2*.

To assess the significance of the fingerprinting analysis, we performed nonparametric permutation testing as in *Finn et al., 2015*. Briefly, the described fingerprinting analysis was repeated for all scans and database-target combinations, but for the identification test, the subject identity in the database set was permuted. In this way, a 'successful' identification was designated when the highest correlation coefficient was between the FC matrices of two different subjects. As expected, for all the nuisance and 'neural' datasets, the identification accuracy estimated with nonparametric permutation testing was around 0.3%, which corresponds to the probability of selecting a specific subject from a group of 392 subjects when the subject is selected at random.

## Acknowledgements

This work was supported by the Natural Sciences and Engineering Research Council of Canada (Discovery Grant 34362 awarded to GDM), the Fonds de la Recherche du Quebec - Nature et Technologies (FRQNT; Team Grant PR191780-2016 awarded to GDM) and the Canada First Research Excellence Fund (awarded to McGill University for the Healthy Brains for Healthy Lives initiative). AXP and MK acknowledge funding from Québec Bio-imaging Network (QBIN). Data were provided by the Human Connectome Project, WU-Minn Consortium (Principal Investigators: David Van Essen and Kamil Ugurbil; 1U54MH091657) funded by the 16 NIH Institutes and Centers that support the NIH Blueprint for Neuroscience Research; and by the McDonnell Center for Systems Neuroscience at Washington University. We are extremely grateful to all Human Connectome Project study participants, who generously donated their time to make this resource possible.

## Additional information

### Funding

| Funder | Grant reference number | Author |
|---|---|---|
| Natural Sciences and Engineering Research Council of Canada | Discovery Grant 34362 | Georgios D Mitsis |
| Fonds de Recherche du Québec - Nature et Technologies | PR191780-2016 | Georgios D Mitsis |
| Canada First Research Excellence Fund | Healthy Brains for Healthy Lives Scholarship | Michalis Kassinopoulos |
| Quebec Bio-Imaging Network | Doctoral Scholarships | Alba Xifra-Porxas Michalis Kassinopoulos |

The funders had no role in study design, data collection and interpretation, or the decision to submit the work for publication.

### Author contributions

Alba Xifra-Porxas, Conceptualization, Data curation, Software, Formal analysis, Investigation, Visualization, Methodology, Writing - original draft, Writing - review and editing; Michalis Kassinopoulos, Conceptualization, Data curation, Investigation, Methodology, Writing - review and editing; Georgios D Mitsis, Conceptualization, Supervision, Funding acquisition, Writing - review and editing

### Author ORCIDs

Alba Xifra-Porxas  https://orcid.org/0000-0002-9023-2432
Michalis Kassinopoulos  http://orcid.org/0000-0003-4312-4401

### Ethics

Human subjects: Human subjects: HCP data were acquired using protocols approved by the Washington University Institutional Review Board (Mapping the Human Connectome: Structure, Function, and Heritability; IRB # 201204036). Informed consent was obtained from subjects. Anonymised data are publicly available from ConnectomeDB (db.humanconnectome.org).

### Decision letter and Author response

Decision letter https://doi.org/10.7554/eLife.62324.sa1
Author response https://doi.org/10.7554/eLife.62324.sa2

## Additional files

### Supplementary files

• Transparent reporting form

## Data availability

Matlab code used in this work can be found here: https://github.com/axifra/Nuisance_signatures_FC (copy archived at https://archive.softwareheritage.org/swh:1:rev: 52781e743d4b4eb491b9330210dac52dcd46fd10).

The following previously published dataset was used:

| Author(s) | Year | Dataset title | Dataset URL | Database and Identifier |
|---|---|---|---|---|
| Essen DV, Ugurbil K | 2017 | Human Connectome Project: WU-Minn HCP consortium | https://www.humanconnectome.org/study/hcp-young-adult | Young Adult, S1200 |

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
