## [Decision Letter]

**Acceptance summary:**

This paper develops a new approach to disentangle various noise contributions, arising from different sources of non-neuronal physiological fluctuations and head motion, to functional magnetic resonance imaging signals. This work also shows that controlling for these different sources of noise improves the accuracy with which the imaging signals can be used to identify people, demonstrating that neuronal activity makes a large contribution to individual differences in patterns of inter-regional functional coupling.

**Decision letter after peer review:**

Thank you for submitting your article "Physiological and motion signatures in static and time-varying functional connectivity and their subject identifiability" for consideration by *eLife*. Your article has been reviewed by 2 peer reviewers, one of whom is a member of our Board of Reviewing Editors, and the evaluation has been overseen by Timothy Behrens as the Senior Editor. The reviewers have opted to remain anonymous.

The reviewers have discussed the reviews with one another and the Reviewing Editor has drafted this decision to help you prepare a revised submission.

Summary:

Both reviewers agree that the manuscript is well-written, thorough, timely, and useful. The authors use various motion and physiological traces to generate surrogate datasets reflecting their contribution to network properties and evaluate how these surrogate data compare to the empirical BOLD network properties measured after different levels of denoising. A strength is the systematic characterization of major physiological and head motion effects, along with widely used fMRI pre-processing pipelines, using a large set of subjects from HCP. While it is generally hard to tell whether a pipeline is reducing nuisance effects without also removing neural signal, consideration of both (i) similarity to nuisance patterns, and (ii) changes in fingerprinting accuracy present informative metrics.

Essential Revisions:

A limitation of this work is that the contributions of various noise sources are determined using zero-lag correlations between motion/physiological and BOLD time series. It is well documented that fluctuations in respiration and motion lead to delayed changes in the BOLD signal that can occur several TRs later, and which can persist for periods that extend beyond the duration of the physiological/motion event (Power et al., NeuroImage, 2015; Power et al., NeuroImage, 2017). This has limited the efficacy of traditional regression-based denoising methods such as RETROICOR (Power et al., NeuroImage, 2017). While the authors have developed a method for convolving these signals with appropriate response functions to obtain the SLFO measure, it is difficult to determine the degree to which this signal captures the key phenomena of interest. The same can be said for motion estimates. It would be very helpful to see examples of individual subjects, with key nuisance traces shown above carpet plots. This would allow readers to ascertain whether the proposed approach accurately captures noise contributions to the BOLD signal.

For evaluating preprocessing pipelines, the primary metric used here is the (dis)similarity between the resulting correlation matrices and those of the nuisance (physio, motion) profiles. Although this is a reasonable approach, it is not clear whether it leads to a more accurate quantification of neural patterns, as the authors also acknowledge. Support is provided by examining the associated fMRI fingerprinting accuracy. A complementary approach, which may further strengthen the claim, could be to compare the post-correction matrices of a high-motion (or high physio) subset of subjects against the raw FC matrices of a subset of subjects that had low motion (or physio) effects to begin with.

To test the significance of each nuisance process in its contribution to BOLD, a surrogate dataset was constructed by permuting the nuisance signals across subjects. It seems like the shuffling procedure was only performed once (for a given nuisance process), and a t-test was performed between the permuted and actual values within each brain region. It may help to shuffle multiple times and pool the results to construct null distributions.

Moreover, the method for generating surrogate data adds Gaussian noise to the estimated nuisance signal contributions. Gaussian noise is not a realistic benchmark for fMRI data and is likely to under-estimate the correlation between the surrogate and empirical data. An autocorrelated process may be a more appropriate choice here.

Could the addition of model-based regressors help to reduce physiological effects in the FCD analysis? In addition, while the FCD analysis focuses on pairwise correlations between time-windowed patterns, it doesn't consider how the patterns themselves are changing as a result of different processing steps. The authors might consider some analysis of the windowed FC patterns, such as summary metrics of their similarity to SLFO profiles.

The Discussion (3.4) mentions that overall, the benefits of GSR may outweigh the possible loss of neuronal signal. Although GSR improves connectome fingerprinting for several of the pipelines, the Results mention that GSR produces negative correlations with the SLFO profile for some scans. I might suspect that for scans in which SLFOs contribute strongly, GSR can help; whereas for scans in which physio is fairly constant over time, GSR is more likely to remove neuronal signal or induce artificial negative correlations, since the GS would contain a larger proportion of neural BOLD. I'd suggest including some discussion of these points in section 3.4.

While GSR is shown to substantially reduce SLFO effects, it has been shown that different brain areas have heterogenous responses to low-frequency physiology (e.g. JE Chen et al. NI 2020), suggesting that a single global regressor may not be the most effective. The authors may wish to provide some discussion of this point in the context of the current findings.

It is noted (p. 9) that GSR tended to cause more negative correlations when performed in volumetric, as opposed to surface, space. It would be helpful to provide some discussion about why this may be the case.

There are 25 regressors in the noise model. To what extent are these collinear?

---

## [Author Response]

Essential Revisions:A limitation of this work is that the contributions of various noise sources are determined using zero-lag correlations between motion/physiological and BOLD time series. It is well documented that fluctuations in respiration and motion lead to delayed changes in the BOLD signal that can occur several TRs later, and which can persist for periods that extend beyond the duration of the physiological/motion event (Power et al., NeuroImage, 2015; Power et al., NeuroImage, 2017). This has limited the efficacy of traditional regression-based denoising methods such as RETROICOR (Power et al., NeuroImage, 2017). While the authors have developed a method for convolving these signals with appropriate response functions to obtain the SLFO measure, it is difficult to determine the degree to which this signal captures the key phenomena of interest. The same can be said for motion estimates. It would be very helpful to see examples of individual subjects, with key nuisance traces shown above carpet plots. This would allow readers to ascertain whether the proposed approach accurately captures noise contributions to the BOLD signal.

In the present work, we used model-based techniques, such as RETROICOR, to identify physiological and head motion signatures in static and time-varying functional connectivity (FC). Model-based techniques that rely on simultaneous physiological recordings or motion realignment parameters are in principle the best way to disentangle the effects of individual nuisance processes (i.e. SLFOs, head motion, breathing motion, cardiac pulsatility) on FC in a quantitative manner. Despite their limitations, model-based techniques employed in the present work explain a significant fraction of variance at the individual level in areas that are known to be affected by the examined nuisance processes (updated Figure 1 in manuscript). As suggested, we have included in the revised manuscript carpet plots with nuisance traces for three subjects (added in manuscript as Figure 7 —figure supplement 3) where it can be appreciated that the strong fluctuations observed in the carpet plot and the global signal are explained to a large degree by SLFOs. Note that SLFOs are derived by convolution models that account for delayed effects up to ~30 seconds, such as those induced by deep breaths and abrupt changes in heart rate. Carpet plots and key nuisance traces for all subjects are available on figshare (Xifra-Porxas et al., 2021).

The following sentences have been added in Methods (section 5.3 – Nuisance processes evaluated):

“Physiological and motion traces from three illustrative scans, along with fMRI carpet plots, can be found in Figure 7 —figure supplement 3. Nuisance traces and carpet plots from all subjects are available on figshare (Xifra-Porxas et al., 2021).” (page 33)

For evaluating preprocessing pipelines, the primary metric used here is the (dis)similarity between the resulting correlation matrices and those of the nuisance (physio, motion) profiles. Although this is a reasonable approach, it is not clear whether it leads to a more accurate quantification of neural patterns, as the authors also acknowledge. Support is provided by examining the associated fMRI fingerprinting accuracy. A complementary approach, which may further strengthen the claim, could be to compare the post-correction matrices of a high-motion (or high physio) subset of subjects against the raw FC matrices of a subset of subjects that had low motion (or physio) effects to begin with.

We agree that it would be reasonable to expect that the FC matrices of a high-motion (high-physio) subset of subjects would look more similar to the matrices of a low-motion (low-physio) subset after being corrected for motion (physiological) artifacts. However, the data-driven preprocessing strategies examined in the present work target both motion and physiologically-related artifacts. As a result, if high-motion subjects are preprocessed with a specific pipeline (e.g. FIX), the contribution of physiological processes (e.g. SLFOs) would also be expected to be lower compared to the low-motion subset. Therefore, it is unclear whether the FC matrices of the two subsets should be expected to look more similar. The same applies to the case of low- and high-physio subsets.

To address the aforementioned issue, we would need to compare preprocessed FC matrices of high-motion or high-physio subjects to the raw FC matrices of subjects who were not affected by neither motion nor physiological artifacts. However, based on the framewise displacement (FD) and the contributions of SLFOs on the global signal, only in a small fraction of scans the effects of both motion and physiological processes on the BOLD signal can be considered negligible (Author response image 1), which does not allow us to perform a reliable comparison. Therefore, to complement fingerprinting accuracy, a more sophisticated approach is required to assess whether the examined preprocessing pipelines lead to more accurate measures of neural patterns, which was out of the scope of the current investigation.

**Author response image 1. sa2fig1:** Plot of mean framewise displacement (FD), which is an estimate of how much a subject moved inside the scanner, versus the correlation between the global signal (GS) and SLFOs, which is an estimate of how much physiological processes influenced the BOLD signal. Each dot represents a scan.Most scans were low motion but were considerably affected by physiological processes.

To test the significance of each nuisance process in its contribution to BOLD, a surrogate dataset was constructed by permuting the nuisance signals across subjects. It seems like the shuffling procedure was only performed once (for a given nuisance process), and a t-test was performed between the permuted and actual values within each brain region. It may help to shuffle multiple times and pool the results to construct null distributions.

As suggested, we generated 1,000 surrogate datasets, where each surrogate dataset consisted of permuting the nuisance signals across subjects. A null distribution for each brain region was computed by estimating the mean contribution to BOLD across subjects for each surrogate nuisance dataset, leading to a distribution of 1,000 values. Subsequently, the significance of an observed nuisance contribution to BOLD was assessed by comparing the observed contribution (mean across subjects) to this null distribution. Multiple comparisons were taken into account by controlling the false discovery rate (FDR). The results of this analysis can be seen in updated Figure 1 in the revised manuscript. The spatial patterns are almost identical to the ones previously reported; however, note that fewer brain regions exhibited a significant contribution of cardiac pulsatility to the BOLD signal.

The relevant section in the Methods (section 5.7 – Statistics) was updated as follows:

“To assess the significance of the results, surrogate nuisance datasets were generated via inter-subject surrogates (Lancaster et al., 2018), using fMRI data recorded from one subject’s scan and physiological signals recorded from a different subject’s scan (in the case of the head motion dataset, volume realignment parameters were employed). Note that when creating each surrogate dataset, only the nuisance process being examined was replaced by signals from a different subject, whereas all other nuisance regressors remained the same. This procedure was repeated 1,000 times for each nuisance process, where each surrogate dataset consisted of permuting the nuisance signals across subjects. A null distribution for each brain region was computed by estimating the mean contribution to BOLD across subjects for each surrogate nuisance dataset, leading to a distribution of 1,000 values. Subsequently, the significance of the nuisance contributions to the BOLD signal were assessed by comparing the observed contribution (mean across subjects) to the corresponding null distribution for each ROI. The significance level at p<0.05 was corrected for multiple comparisons using false discovery rate (FDR). Furthermore, the similarity between the nuisance and “neural” FC matrices was compared against the similarity obtained using surrogate nuisance FC matrices.” (page 39)

Moreover, the method for generating surrogate data adds Gaussian noise to the estimated nuisance signal contributions. Gaussian noise is not a realistic benchmark for fMRI data and is likely to under-estimate the correlation between the surrogate and empirical data. An autocorrelated process may be a more appropriate choice here.

Thank you for this suggestion. We generated the surrogate datasets again using an autocorrelated process as noise. We modelled the autocorrelated process as a first order autoregressive (AR(1)) model: ψ(t)=α1ψ(t−1)+ξ(t)

We used a first order AR model as it has been shown to be able to capture both the static and dynamic FC structure of resting-state fMRI data (Liégeois et al., 2017). For each ROI in the surrogate datasets, the coefficient 𝑎1 was randomly sampled from a distribution of coefficients generated through fitting an AR(1) model to the real fMRI data (added to manuscript as Figure 7 —figure supplement 5). The results using autocorrelated noise were very similar to the previously reported findings using Gaussian noise, as can be seen for the static FC results in Author response image 2. All figures in the manuscript were updated accordingly, whereas the main findings remained the same.

The relevant section in Methods (section 5.4 – Isolation of nuisance fluctuations from fMRI data) was updated as follows:

Afterwards, nuisance datasets for each process were created by scaling the estimated nuisance signal within each ROI with its corresponding correlation coefficient 𝑟_𝑛𝑢𝑖𝑠_ and adding a first order autoregressive (AR(1)) process (ѱ(t)) scaled with 𝑟_𝑛𝑒𝑢𝑟_ . We used an AR(1) model as it has been shown to be able to capture both the static and time-varying FC structure of resting-state fMRI data (Liégeois et al., 2017). This is expressed as:ZyNuis(t)=rnuis[ŷNPI(t)]+rneurZ[ψ(t)]where 𝑍[∙] denotes normalization to zero mean and unit variance. The coefficient α_1_ was randomly sampled from a distribution of coefficients generated through fitting an AR(1) model to the real fMRI data (Figure 7 – figure supplement 5).” (pages 34-35).

**Author response image 2. sa2fig2:** Effect of preprocessing strategies using(A) Gaussian noise or(B) autocorrelated noise to generate the surrogate datasets.

Could the addition of model-based regressors help to reduce physiological effects in the FCD analysis? In addition, while the FCD analysis focuses on pairwise correlations between time-windowed patterns, it doesn't consider how the patterns themselves are changing as a result of different processing steps. The authors might consider some analysis of the windowed FC patterns, such as summary metrics of their similarity to SLFO profiles.

We assessed the potential benefit of model-based regressors in reducing physiological effects in the FCD analysis, and even though there was some reduction in the effects of SLFOs on neural FCD matrices, the reductions were not large enough for the similarity values to reach chance levels (added to manuscript as Figure 5 —figure supplement 1). We have added the following paragraphs to Results (section 2.5):

“We observed that the temporal evolution of FC patterns from SLFOs and head motion were similar to the ones observed in the raw data. An illustration of this similarity is shown in Figure 6 for six subjects. On the other hand, the distribution of similarity values for breathing motion and cardiac pulsatility FCD matrices was around zero, indicating the absence of systematic effects on “neural” FCD matrices (Figure 5). Including model-based regressors in the preprocessing pipelines led to a small decrease in the similarity between neural and SLFOs FCD matrices, particularly for mild pipelines and FIX without GSR, even though these decreased similarity values were still above chance levels (Figure 5 —figure supplement 1A).” (page 15)

Further, we modified the following sentence in Discussion (section 3.3):

“Importantly, neither data-driven nor model-based preprocessing strategies were able to completely remove these confounds (Figure 5 —figure supplement 1).” (page 23)

With regards to the effects of physiological processes on time-varying FC, as suggested by the Reviewers, we assessed the similarity of the “neural” signatures after each preprocessing pipeline to the nuisance signatures (e.g. SLFOs signature) on a window-by-window basis. Specifically, for a given scan, we first computed the similarity (i.e. Pearson correlation coefficient) between the “neural” and nuisance signature on a window-by-window basis and, subsequently, averaged the similarity values across windows. Author response image 3 shows the distribution of the mean similarity values across all 1,568 scans for all preprocessing strategies and nuisance processes. As can be observed, the results of the window-based analysis are very similar to the static FC analysis (Figure 2E-H), with the main difference being that the similarity values were lower in the window-based analysis, likely due to the estimation of FC matrices from shorter duration recordings (43.2 sec vs 835.2 sec).

We did not include the results of this analysis in the revised manuscript, as it likely reflects the degree to which a nuisance process can affect FC estimates overall within a scan, regardless of whether the associated effect is constant or partly accounts for time-varying FC estimates. In contrast, the FCD analysis considered in the present work quantifies the degree to which the recurrent patterns observed in the “neural” datasets could be attributed to fluctuations in physiological processes.

**Author response image 3. sa2fig3:** Similarity between “neural” and nuisance FC patterns within shorter windows. Distribution of similarity (Pearson correlation coefficient) values between “neural” time-windowed FC matrices applying different preprocessing strategies and nuisance time-windowed FC matrices associated to (A)SLFOs, (B)head motion, (C)breathing motion, and (D)cardiac pulsatility. Correlation values were Fisher z transformed.

The Discussion (3.4) mentions that overall, the benefits of GSR may outweigh the possible loss of neuronal signal. Although GSR improves connectome fingerprinting for several of the pipelines, the Results mention that GSR produces negative correlations with the SLFO profile for some scans. I might suspect that for scans in which SLFOs contribute strongly, GSR can help; whereas for scans in which physio is fairly constant over time, GSR is more likely to remove neuronal signal or induce artificial negative correlations, since the GS would contain a larger proportion of neural BOLD. I'd suggest including some discussion of these points in section 3.4.While GSR is shown to substantially reduce SLFO effects, it has been shown that different brain areas have heterogenous responses to low-frequency physiology (e.g. JE Chen et al. NI 2020), suggesting that a single global regressor may not be the most effective. The authors may wish to provide some discussion of this point in the context of the current findings.

We thank the Reviewers for their valuable comment. We further investigated why GSR produces negative correlations between the SLFOs signatures and neural FC matrices for some scans, and found that the scans that exhibited this behavior were scans where a large fraction of the GS variance was attributed to SLFOs (Figure 2 —figure supplement 4A). In contrast, scans with low correlation between GS and SLFOs, which are likely scans with fairly constant cardiac and breathing rhythms, were not substantially impacted by GSR. In addition, we observed that for scans with negative SLFOs-neural similarity after GSR, the estimated SLFOs signature yielded relatively low contributions in the somatosensory and auditory networks compared to other networks (Figure 2 —figure supplement 4B, left), while these two networks exhibited high correlation values in the case of neural signatures after GSR (Figure 2 —figure supplement 4B, right). In contrast, scans whose SLFOs-neural similarity was close to zero after GSR (Figure 2 —figure supplement 4C) did not exhibit increased FC in the somatosensory and auditory networks (Figure 2 —figure supplement 4D, right). Thus, the different levels of correlation in these two networks compared to other networks are likely responsible for the negative SLFOs-neural similarity after GSR.

A possible explanation for the low contribution of SLFOs in the somatosensory and auditory networks is the use of a single global regressor for the estimation of the SLFOs signature, which does not account for variability in the dynamics of SLFOs across brain regions. Interestingly, the primary sensory areas that belong to the two aforementioned networks were also reported in the work by Chen et al. (2020) as exhibiting a different respiration response function (RRF) compared to other areas. These findings suggest that the modelling of SLFOs at the individual level could potentially be improved if spatial variability in the RRF is also accounted for. Moreover, these results indicate that the effects of SLFOs cannot be corrected equally well in all areas solely by regressing out the GS. It should be noted, though, that the spatial variability of RRF reported in Chen et al. (2020) was observed at the group level using a relatively flexible model with ten free parameters. Future studies are still needed to address how the SLFOs can be modelled at the individual level in a manner that takes into consideration both subject and spatial specificity, while avoiding overfitting.

We would like to clarify that the negative SLFOs-neural similarity after GSR is not related to the overall decrease in connectivity strength observed in the neural signatures (“inflation” of negative correlations) rather than to the change of the connectome pattern in the neural signature (i.e. some edges are affected by GSR to a larger extent than other edges). In addition, we would like to point out that the negative SLFOs-neural similarity was substantially decreased when GSR is combined with white matter denoising, which may suggest that the nuisance regressors obtained from the white matter can reduce fluctuations induced by SLFOs in all areas, including primary sensory areas, better than GSR alone.

Figure 2 —figure supplement 4 has been added to the revised manuscript and the two following paragraphs with regards to the findings of this secondary analysis have been added to the discussion (section 3.4):

“For some scans, GSR led to a negative similarity between the SLFOs and neural signatures (Figure 2E). Further investigation revealed that scans that exhibited a negative SLFOs-neural similarity after GSR were scans where a large fraction of the GS variance was attributed to SLFOs (Figure 2 —figure supplement 4A). In contrast, scans with low correlation between GS and SLFOs, which are likely scans with fairly constant cardiac and breathing rhythms, were not affected by GSR. In addition, we observed that in the case of scans with negative SLFOs-neural similarity after GSR, the estimated SLFOs signature yielded relatively low contributions to the somatosensory and auditory network compared to other networks (Figure 2 —figure supplement 4B, left), while these two networks exhibited high correlation values for the neural signature after GSR (Figure 2 —figure supplement 4B, right). In contrast, scans whose SLFOs-neural similarity was close to zero after GSR (Figure 2 —figure supplement 4C) did not exhibit increased FC in the somatosensory and auditory networks (Figure 2 —figure supplement 4D). Thus, the different levels of correlation in these two networks, as compared to other networks, were likely responsible for the negative GS-SLFOs similarity after GSR. A possible explanation for the low contribution of SLFOs in the somatosensory and auditory networks is the use of a single global regressor for the estimation of the SLFOs signature, which does not account for variability in the dynamics of SLFOs across brain regions. Interestingly, the primary sensory areas that belong to the two aforementioned networks were also reported in the work by Chen et al., (2020) as exhibiting a different respiration response function (RRF) compared to other areas. These findings suggest that modelling SLFOs at the individual level could potentially be improved if spatial variability in RRF is also accounted for. Moreover, these results indicate that the effects of SLFOs cannot be corrected equally well in all areas solely by regressing out the GS. It should be noted, though, that the RRF spatial variability reported by Chen et al., (2020) was observed at the group level using a relatively flexible model with ten free parameters. Future studies are still needed to address how the SLFOs can be modelled at the individual level in a manner that takes into consideration both subject and spatial specificity, while avoiding overfitting. Related to this, we also observed that the negative SLFOs-neural similarity was substantially decreased when GSR is combined with white matter denoising, which may suggest that the nuisance regressors obtained from white matter can reduce fluctuations induced by SLFOs in all areas, including primary sensory areas, better than GSR alone.” (page 24)

It is noted (p. 9) that GSR tended to cause more negative correlations when performed in volumetric, as opposed to surface, space. It would be helpful to provide some discussion about why this may be the case.

Unfortunately, we do not have a clear explanation for this effect. We feel that any discussion we could provide would be highly speculative, and thus we prefer to just report this observation.

There are 25 regressors in the noise model. To what extent are these collinear?

The 25 regressors included in the model are not collinear (Figure supplement 4 for Figure 7). The highest correlations were between realignment parameters and their derivatives, but even those were below 0.5.

The following sentences have been added in Methods (section 5.4 – Isolation of nuisance fluctuations from fMRI data):

“Note that no significant collinearity was observed between the 25 regressors (Figure 7 —figure supplement 4).” (page 33)